# DIGNet: Learning Decomposed Patterns in Representation Balancing for Treatment Effect Estimation

**Yiyan Huang** [*]                                            *yiyhuang@polyu.edu.hk*
*Department of Applied Mathematics, The Hong Kong Polytechnic University*

**Siyi Wang** [*]                                              *siyi.wang@my.cityu.edu.hk*
*School of Data Science, City University of Hong Kong*

**Cheuk Hang Leung**                                           *chleung87@cityu.edu.hk*
*School of Data Science, City University of Hong Kong*

**Qi Wu** [†]                                                  *qi.wu@cityu.edu.hk*
*School of Data Science, City University of Hong Kong*

**Dongdong Wang**                                              *wangdongdong9@jd.com*
*JD Digits*

**Zhixiang Huang**                                             *huangzhixiang@jd.com*
*JD Digits*

**Reviewed on OpenReview:** *https://openreview.net/forum?id=Z20FInfWlm*

## Abstract

Estimating treatment effects from observational data is often subject to a covariate shift problem incurred by selection bias. Recent research has sought to mitigate this problem by leveraging representation balancing methods that aim to extract balancing patterns from observational data and utilize them for outcome prediction. The underlying theoretical rationale is that minimizing the unobserved counterfactual error can be achieved through two principles: (I) reducing the risk associated with predicting factual outcomes and (II) mitigating the distributional discrepancy between the treated and controlled samples. However, an inherent trade-off between the two principles can lead to a potential loss of information useful for factual outcome predictions and, consequently, deteriorating treatment effect estimations. In this paper, we propose a novel representation balancing model, DIGNet, for treatment effect estimation. DIGNet incorporates two key components, PDIG and PPBR, which effectively mitigate the trade-off problem by improving one aforementioned principle without sacrificing the other. Specifically, PDIG captures more effective balancing patterns (Principle II) without affecting factual outcome predictions (Principle I), while PPBR enhances factual outcome prediction (Principle I) without affecting the learning of balancing patterns (Principle II). The ablation studies verify the effectiveness of PDIG and PPBR in improving treatment effect estimation, and experimental results on benchmark datasets demonstrate the superior performance of our DIGNet model compared to baseline models.

## 1 Introduction

In the context of the ubiquity of personalized decision-making, causal inference has sparked a surge of research exploring causal machine learning in many disciplines, including economics and statistics (Wager &

---

[*]The co-first authors.

[†]The corresponding author.

Athey, 2018; Athey & Wager, 2019; Farrell, 2015; Chernozhukov et al., 2018; Huang et al., 2021), healthcare (Qian et al., 2021; Bica et al., 2021a;b), and commercial applications (Guo et al., 2020a;b; Chu et al., 2021). The core of causal inference is to estimate *treatment effects*, which is closely related to the *factual outcomes* (observed outcomes) and *counterfactual outcomes*. The concept of the counterfactual outcome is closely linked to a fundamental hypothetical question: What would the outcome be if an alternative treatment were received? Answering this question is challenging because counterfactual outcomes are unobservable in reality, making it impossible to directly access ground-truth treatment effects from observational data. Consequently, an increasing amount of recent research has focused on developing innovative machine learning models that aim to enhance the estimation of counterfactual outcomes to obtain more accurate treatment effect estimates.

One of the challenges in estimating counterfactual outcomes lies in the *covariate shift* problem. In observational data, the population can be typically divided into two groups: (i) individuals who received treatment ($T = 1$), referred to as *treated samples* or *treatment samples*, and (ii) individuals who did not receive treatment ($T = 0$), referred to as *controlled samples* or *control samples*. The covariate shift problem indicates the difference between the distribution of covariate in the treated group and that in the controlled group, meaning $P(X|T = 1) \neq P(X|T = 0)$. This phenomenon is a result of the non-random treatment assignment mechanism, where the decision to receive treatment (e.g., heart medicine) is often determined by the covariate (e.g., age). For example, people receiving heart medicine treatment tend to be much older compared to those who do not receive such treatment, because the doctor's decision-making regarding whether to undergo heart medicine treatment highly depends on the patients' age. Such a non-random treatment assignment is known as the selection bias phenomenon in the causal inference literature.

Although the covariate shift arises from the association between covariate and treatment, this issue can significantly exacerbate the difficulty in inferring counterfactual outcomes, as traditional machine learning models can be invalid in estimating potential outcomes when a covariate shift is present (Yao et al., 2018; Hassanpour & Greiner, 2019a). Specifically, to infer the potential outcome $Y^0$ for treated ($T = 1$) samples, the conventional approach is to first train a model $\hat{\tau}^0(X)$ using controlled ($T = 0$) samples, and then utilize $\hat{\tau}^0(X)$ to predict $Y^0$ for treated ($T = 1$) samples. This approach, known as the T-learner in the causal inference literature (Curth & Van Der Schaar, 2023; Mahajan et al., 2024), becomes problematic because the training data (control samples) used for model training do not have the same distribution as the test data (treated samples), i.e., $P(X|T = 1) \neq P(X|T = 0)$. This violates the assumption in machine learning that training data and test data should be independent and identically distributed.

To alleviate the covariate shift problem, recent advancements in representation balancing research have explored the representation learning model, such as CounterFactual Regression Network (CFRNet) (Shalit et al., 2017), to estimate individual treatment effects (ITEs). These representation balancing models aim to extract balancing patterns from observational data and utilize these patterns to predict outcomes. The corresponding objective function is typically concerned with minimizing the empirical risk of factual outcomes while concurrently minimizing the distributional distance between the treatment and control groups in the representation space (Shalit et al., 2017; Johansson et al., 2022a). The underlying theoretical logic behind these studies is that minimizing counterfactual error can be achieved by two principles in the representation space: *(Principle I) minimizing the risk associated with factual outcome prediction*, and *(Principle II) reducing the distributional discrepancy between the treated and controlled samples*.

While the representation balancing framework provides a powerful tool to address the covariate shift issue, models based on classical structures such as CFRNet (Figure 1(a)) still encounter a trade-off between the aforementioned two principles: Enforcing models to focus solely on balancing can undermine the predictive power of the outcome function (Zhang et al., 2020; Assaad et al., 2021; Huang et al., 2023). A detailed discussion of this trade-off problem can be found in Appendix A.1. This inherent trade-off motivates us to explore a pivotal question: considering the inherent trade-off between the two principles, *is it possible to explore a scheme that enhances one principle without sacrificing the other?* More specifically, can we explore improving treatment effect estimation through the following two paths: *(Path I) learning more effective balancing patterns without sacrificing factual outcome prediction* and *(Path II) enhancing factual outcome prediction without sacrificing the learning of balancing patterns?* In the following, we present the proposed solutions and the rationale behind the underlying intuitions.

In classic representation balancing models, the process of learning balancing patterns can lead to the loss of outcome-predictive information. It is therefore natural to consider a module that can preserve the pre-balancing information before the representation balancing step. This increases the model's complexity to maintain the useful predictive knowledge while still benefiting from the covariate balancing properties of the representation balancing framework. Furthermore, in multi-task learning, distinct representations are learned for different tasks, with each task involving its own objective function. An important step in multi-task learning is integrating the information from these separately learned representations into a unified representation. Therefore, following the multi-task learning paradigm (Li et al., 2018; Baltrušaitis et al., 2018; Crawshaw, 2020; Yan et al., 2021; Xu et al., 2023b), we propose concatenating the representations learned with Wasserstein distance and $\mathcal{H}$-divergence to form a joint representation. This joint representation can effectively capture the task-specific balancing information provided by each distance metric without adversely affecting the outcome modeling task. A detailed discussion of these intuitions can be found in Section 5.3.1.

Based on the above motivations, we introduce a novel representation balancing model, **DIGNet** (Section 5.2.2), a neural network that incorporates ***D**ecomposed patterns* with ***I**ndividual propensity confusion* and ***G**roup distance minimization*. Decomposed patterns denote distinct components disentangled from specific representations in DIGNet (Section 5.2). The individual propensity confusion aspect of DIGNet aims to learn representations that obscure the propensity of each individual being treated or controlled (Section 5.1.2), grounded in our derived $\mathcal{H}$-divergence guided counterfactual and ITE error bounds (Section 4.2). The group distance minimization aspect focuses on learning representations that minimize the distance between treated and controlled groups (Section 5.1.1), supported by previous work on Wasserstein distance guided counterfactual and ITE error bounds (Shalit et al., 2017) (Section 4.1). Figure 1 visually depicts these introduced concepts and their relationships.

**Contributions.**  Our main contributions are summarized as follows:

1. We derive theoretical upper bounds for counterfactual error and ITE error based on $\mathcal{H}$-divergence (Section 4.2). In particular, this theoretical foundation highlights the important role of propensity score for representation balancing models, connecting the representation balancing with the concept of individual propensity confusion.

2. We suggest learning decomposed patterns in representation balancing models (Section 5.2.1) to mitigate the trade-off problem rooted in classic causal representation balancing models. First, we propose a ***PDIG*** method (Figure 1(b)), which aims to learn **P**atterns **D**ecomposed with **I**ndividual propensity confusion and **G**roup distance minimization to improve treatment effect estimation through Path I. Second, we propose a ***PPBR*** method (Figure 1(c)), which aims to learn **P**atterns of **P**re-balancing and **B**alancing **R**epresentations to improve treatment effect estimation through Path II.

3. Building upon PDIG and PPBR, we propose a novel representation balancing model, DIGNet (Figure 1(d)), for treatment effect estimation. In Section 6, ablation studies verify the efficacy of PDIG and PPBR in improving ITE estimation through Path I and Path II, respectively. Furthermore, experimental results on benchmark datasets demonstrate that DIGNet surpasses the performance of baseline models in terms of treatment effect estimation.

## 2 Related Work

The presence of a covariate shift problem stimulates the line of representation balancing works (Johansson et al., 2016; Shalit et al., 2017; Johansson et al., 2022a). These works aim to balance the distributions of representations between treated and controlled groups and simultaneously try to maintain representations predictive of factual outcomes. This idea is closely connected with domain adaptation. In particular, the ITE error bound based on Wasserstein distance is similar to the generalization bound in Ben-David et al. (2010); Long et al. (2014); Shen et al. (2018). The theoretical foundation and the classic CFRNet structure proposed in Shalit et al. (2017) have inspired many subsequent studies on representation balancing methods for treatment effect estimation, including Yao et al. (2018); Shi et al. (2019); Zhang et al. (2020); Hassanpour

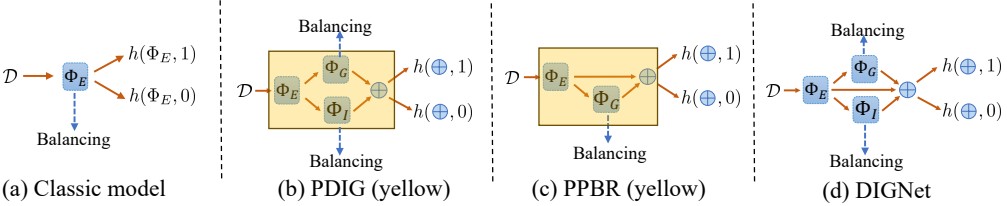

Figure 1: (a): The classic model (e.g., GNet in Section 5.1.1 and INet in Section 5.1.2) maps the original data $\mathcal{D}$ into representations $\Phi_E$ to achieve representation balancing. The balanced representations are referred to as *balancing patterns*. These balancing patterns are also used for outcome prediction. (b): The PDIG (Section 5.2.1) is illustrated as the yellow part, where balancing patterns are decomposed into two distinct components, $\Phi_G$ and $\Phi_I$. $\Phi_G$ serves for *group distance minimization* (Section 5.1.1) and $\Phi_I$ serves for *individual propensity confusion* (Section 5.1.2). The balancing patterns $\Phi_G$ and $\Phi_I$ are concatenated for predicting outcomes. (c): The PPBR (Section 5.2.1) is represented by the yellow section, where $\Phi_E$ is used for feature extraction and $\Phi_G$ is used for representation balancing. Here representations are decomposed into *pre-balancing patterns* $\Phi_E$ and balancing patterns $\Phi_G$. $\Phi_E$ and $\Phi_G$ are concatenated for predicting outcomes. (d): The proposed model DIGNet (Section 5.2.2) integrates both PDIG and PPBR. Specifically, DIGNet decomposes balancing patterns into two distinct components, $\Phi_G$ and $\Phi_I$. The outcome predictors are further formed by concatenating $\Phi_G$, $\Phi_I$, and pre-balancing patterns $\Phi_E$.

& Greiner (2019a); Assaad et al. (2021); Huang et al. (2022a). This paper derives a new ITE error bound based on $\mathcal{H}$-divergence (Ben-David et al., 2006; 2010; Ganin et al., 2016). In addition to the connection to domain adaptation, causal representation learning is also linked to the field of fair representation learning, which aims to ensure that machine learning algorithms make fair decisions by learning fair representations. The main goal of these studies is to enforce a classification model to be less sensitive to certain sensitive variables when the representations of different groups are sufficiently similar (Zemel et al., 2013; Edwards & Storkey, 2015; Beutel et al., 2017; Madras et al., 2018; Zhang et al., 2018; Adel et al., 2019; Feng et al., 2019; Zhao et al., 2019a; Zhao & Gordon, 2022). Notably, the original idea of adversarial learned fair representations in Edwards & Storkey (2015) is also motived by the domain adaptation work (Ben-David et al., 2006; 2010; Ganin et al., 2016), sharing a similar motivation to our utilization of INet, which relies on $\mathcal{H}$-divergence guided error bounds for ITE estimation. Moreover, Wasserstein distance has also been employed for learning fair representations in Jiang et al. (2020).

Another recent line of causal representation learning literature investigates efficient neural network structures for treatment effect estimation. Kuang et al. (2017); Hassanpour & Greiner (2019b) extract the original covariates into treatment-specific factors, outcome-specific factors, and confounding factors; X-learner (Künzel et al., 2019) and R-learner (Nie & Wager, 2021) are developed beyond the classic S-learner and T-learner; Curth & van der Schaar (2021) leverage structures for end-to-end learners to counteract the inductive bias towards treatment effect estimation, which is motivated by Makar et al. (2020). There are some other deep neural network models that have been employed in treatment effect estimation Louizos et al. (2017); Yao et al. (2018); Yoon et al. (2018); Shi et al. (2019); Du et al. (2021). To ensure comparability and consistency, we rigorously follow the same framework as these causal inference works. The causal graph in these studies satisfies the standard setup $T \leftarrow X \rightarrow Y$ and $T \rightarrow Y$. Additionally, it is also worth noting that there are many other causal inference works exploring treatment effect estimation under more complex causal graphs. For instance, studies such as Kallus et al. (2019); Jesson et al. (2021); Miao et al. (2023) specifically tackle the treatment effect estimation when unobserved confounders $U$ present. In this case, the causal graph setup extends to $T \leftarrow X \rightarrow Y$, $T \rightarrow Y$, $T \leftarrow U \rightarrow Y$. A recent work (Cao et al., 2023) further expands this static causal graph to a dynamic setting. Moreover, some studies such as Angrist et al. (1996); Burgess et al. (2017); Wu et al. (2022); Yuan et al. (2023) estimate treatment effects with instrumental variables $I$ involved. In this case, there are various causal graph setups such as $T \leftarrow X \rightarrow Y$, $I \rightarrow T \rightarrow Y$, and $T \leftarrow I \rightarrow Y$. More complex causal graph settings (Nogueira et al., 2021; Vowels et al., 2022; Zanga et al.,

2022) have been studied with the development of Directed Graphical Models (Pearl, 2009), which represents another significant research direction known as causal discovery.

Our method is highly motivated by the trade-off problem between outcome prediction and representation balancing. In the causal representation learning literature, a similar trade-off phenomenon has been noticed by Zhang et al. (2020); Assaad et al. (2021); Huang et al. (2022a), where the researchers argue that highly-balanced representations can have adverse effects on outcome modeling. However, the explanations for this phenomenon and its connections with other related literature are not extensively provided in their work. We highlight that the trade-off between outcome prediction and representation balancing is also connected with trade-offs observed in other research domains. In representation balancing models, representation balancing helps improve the model's ability to generalize to counterfactual estimates. However, representation balancing can potentially sacrifice information necessary for predicting factual outcomes. In supervised machine learning, penalizing model complexity during model training helps the model to learn simpler patterns, thereby promoting generalization ability (reducing its variance) to unseen data. However, a bias-variance trade-off occurs because less flexible models tend to exhibit higher bias in training data (Geman et al., 1992; Domingos, 2000; Valentini & Dietterich, 2004; Yang et al., 2020). In the literature of domain adaptation (Shen et al., 2018; Zhao et al., 2019b), transfer learning (Long et al., 2015; 2017; Ma et al., 2023), out-of-distribution detection (Kumar et al., 2021; 2022), and fair representation learning (Zliobaite, 2015; Hardt et al., 2016), enforcing a model to capture proxy features that are domain-invariant helps the model to generalize well to unseen target (also known as out-of-distribution) data. However, a trade-off between classification accuracy and domain-invariance (or fairness in fair representation learning literature) occurs because the pursuit of domain-invariant features may lead to a loss of classification accuracy on the source (also known as in-distribution) data (Zhao et al., 2019a; Zhao & Gordon, 2022; Zhao et al., 2022).

## 3 Preliminaries

**Notations.** Suppose there are $N$ i.i.d. random variables $\mathcal{D} = \{(\mathbf{X}_i, T_i, Y_i)\}_{i=1}^N$ with observed realizations $\{(\mathbf{x}_i, t_i, y_i)\}_{i=1}^N$, where there are $N_1$ treated units and $N_0$ controlled units. For each unit $i$, $\mathbf{X}_i \in \mathcal{X} \subset \mathbb{R}^d$ denotes $d$-dimensional covariates and $T_i \in \{0, 1\}$ denotes the binary treatment, with $e(\mathbf{x}_i) := p(T_i = 1 \mid \mathbf{X}_i = \mathbf{x}_i)$ defined as the propensity score (Rosenbaum & Rubin, 1983). Potential outcome framework (Rubin, 2005) defines the potential outcomes $Y^1, Y^0 \in \mathcal{Y} \subset \mathbb{R}$ for treatment $T = 1$ and $T = 0$, respectively. We let the observed outcome (factual outcome) be $Y = T \cdot Y^1 + (1 - T) \cdot Y^0$, and the unobserved outcome (counterfactual outcome) be $Y = T \cdot Y^0 + (1 - T) \cdot Y^1$. For $t \in \{0, 1\}$, let $\tau^t(\mathbf{x}) := \mathbb{E}[Y^t \mid \mathbf{X} = \mathbf{x}]$ be a function of $Y^t$ w.r.t. $\mathbf{X}$, then our goal is to estimate the individual treatment effect (ITE) $\tau(\mathbf{x}) := \mathbb{E}[Y^1 - Y^0 \mid \mathbf{X} = \mathbf{x}] = \tau^1(\mathbf{x}) - \tau^0(\mathbf{x})$ [1], and the average treatment effect (ATE) $\tau_{ATE} := \mathbb{E}[Y^1 - Y^0] = \int_{\mathcal{X}} \tau(\mathbf{x}) p(\mathbf{x}) d\mathbf{x}$. The introduced concepts PPBR and PDIG are illustrated in Figure 1, and the necessary representation functions $\Phi_E$, $\Phi_G$ and $\Phi_I$, as well as different model structures, are illustrated in Figure 2. Throughout the paper, we refer to patterns as meaningful representations. For instance, decomposed patterns are distinct components disentangled from some specific representations.

### 3.1 Problem setup

In causal representation balancing works, we denote representation space by $\mathcal{R} \subset \mathbb{R}^d$, and $\Phi : \mathcal{X} \to \mathcal{R}$ is assumed to be a twice-differentiable, one-to-one and invertible function with its inverse $\Psi : \mathcal{R} \to \mathcal{X}$ such that $\Psi(\Phi(\mathbf{x})) = \mathbf{x}$ [1]. The densities of the treated and controlled covariates are denoted by $p_{\mathbf{x}}^{T=1} = p^{T=1}(\mathbf{x}) := p(\mathbf{x} \mid T = 1)$ and $p_{\mathbf{x}}^{T=0} = p^{T=0}(\mathbf{x}) := p(\mathbf{x} \mid T = 0)$, respectively. Correspondingly, the densities of the treated and controlled covariates in the representation space are denoted by $p_{\Phi}^{T=1} = p_{\Phi}^{T=1}(\mathbf{r}) := p_{\Phi}(\mathbf{r} \mid T = 1)$ and $p_{\Phi}^{T=0} = p_{\Phi}^{T=0}(\mathbf{r}) := p_{\Phi}(\mathbf{r} \mid T = 0)$, respectively.

---

[1]The term $\mathbb{E}[Y^1 - Y^0 \mid \mathbf{X} = \mathbf{x}]$ is commonly known as the Conditional Average Treatment Effect (CATE). In order to maintain consistency with the notion used in the existing causal representation balancing literature, e.g., Shalit et al. (2017), we refer to this term as ITE throughout this paper. Note that the original definition of ITE for the $i$-th individual is commonly expressed as the difference between their potential outcomes, represented as $Y_i^1 - Y_i^0$.

[1]Theoretically, the invertibility is necessary for deriving the upper bounds of ITE error, specifically for equation 39 and equation 47. However, the invertibility can be hard to verify in practice (Johansson et al., 2022b).

Our study is based on the potential outcome framework (Rubin, 2005). Assumption 1 states standard and necessary assumptions to ensure treatment effects are identifiable. Before proceeding with theoretical analysis, we also present some necessary terms and definitions in Definition 1.

**Assumption 1** (Consistency, Overlap, and Unconfoundedness). *Consistency: If the treatment is $t$, then the observed outcome equals $Y^t$. Overlap: The propensity score is bounded away from 0 to 1, i.e., $0 < e(\mathbf{x}) < 1$. Unconfoundedness: $Y^t \perp\!\!\!\perp T \mid \mathbf{X}, \forall t \in \{0, 1\}$.*

**Definition 1.** *Let $h : \mathcal{R} \times \{0, 1\} \to \mathcal{Y}$ be an hypothesis defined over the representation space $\mathcal{R}$ such that $h(\Phi(\mathbf{x}), t)$ estimates $y^t$, and $L : \mathcal{Y} \times \mathcal{Y} \to \mathbb{R}_+$ be a loss function (e.g., the squared loss $L(y, y') = (y - y')^2$ or the absolute loss $L(y, y') = |y - y'|$). If we define the expected loss for $(\mathbf{x}, t)$ as $\ell_{h,\Phi}(\mathbf{x}, t) = \int_{\mathcal{Y}} L(y^t, h(\Phi(\mathbf{x}), t)) p(y^t|\mathbf{x}) dy^t$, we then have factual and counterfactual errors, as well as them on the treated and controlled:*

$$\epsilon_F(h, \Phi) = \int_{\mathcal{X} \times \{0,1\}} \ell_{h,\Phi}(\mathbf{x}, t) p(\mathbf{x}, t) d\mathbf{x} dt, \qquad \epsilon_{CF}(h, \Phi) = \int_{\mathcal{X} \times \{0,1\}} \ell_{h,\Phi}(\mathbf{x}, t) p(\mathbf{x}, 1 - t) d\mathbf{x} dt,$$

$$\epsilon_F^{T=1}(h, \Phi) = \int_{\mathcal{X}} \ell_{h,\Phi}(\mathbf{x}, 1) p^{T=1}(\mathbf{x}) d\mathbf{x}, \qquad \epsilon_F^{T=0}(h, \Phi) = \int_{\mathcal{X}} \ell_{h,\Phi}(\mathbf{x}, 0) p^{T=0}(\mathbf{x}) d\mathbf{x},$$

$$\epsilon_{CF}^{T=1}(h, \Phi) = \int_{\mathcal{X}} \ell_{h,\Phi}(\mathbf{x}, 1) p^{T=0}(\mathbf{x}) d\mathbf{x}, \qquad \epsilon_{CF}^{T=0}(h, \Phi) = \int_{\mathcal{X}} \ell_{h,\Phi}(\mathbf{x}, 0) p^{T=1}(\mathbf{x}) d\mathbf{x}.$$

If we let $f(\mathbf{x}, t)$ be $h(\Phi(\mathbf{x}), t)$, where $f : \mathcal{X} \times \{0, 1\} \to \mathcal{Y}$ is a prediction function for outcome, then the estimated ITE over $f$ is defined as $\hat{\tau}_f(\mathbf{x}) := f(\mathbf{x}, 1) - f(\mathbf{x}, 0)$. We can measure the error in ITE estimation with the metric, Precision in the expected Estimation of Heterogeneous Effect (PEHE):

$$\epsilon_{PEHE}(f) = \int_{\mathcal{X}} L(\hat{\tau}_f(\mathbf{x}), \tau(\mathbf{x})) p(\mathbf{x}) d\mathbf{x}. \tag{1}$$

Here, $\epsilon_{PEHE}(f)$ can also be denoted by $\epsilon_{PEHE}(h, \Phi)$ if we let $f(\mathbf{x}, t)$ be $h(\Phi(\mathbf{x}), t)$.

## 4    Theoretical Results

In this section, we first prove $\epsilon_{PEHE}$ is bounded by $\epsilon_F$ and $\epsilon_{CF}$ in Lemma 1. Next, we revisit the upper bound for Wasserstein distance guided representation balancing method in Section 4.1. Furthermore, we state the new theoretical results concerning $\mathcal{H}$-divergence guided representation balancing method in Section 4.2.

**Lemma 1.** *Let functions $h$ and $\Phi$ be as defined in Definition 1. Recall that $\tau^t(\mathbf{x}) = \mathbb{E}[Y^t \mid \mathbf{X} = \mathbf{x}]$. Define $\sigma_y^2 = \min\{\sigma_{y^t}^2(p(\mathbf{x}, t)), \sigma_{y^t}^2(p(\mathbf{x}, 1 - t))\}$ and $A_y = \max\{A_{y^t}(p(\mathbf{x}, t)), A_{y^t}(p(\mathbf{x}, 1 - t))\} \; \forall t \in \{0, 1\}$, where $\sigma_{y^t}^2(p(\mathbf{x}, t)) = \int_{\mathcal{X} \times \{0,1\} \times \mathcal{Y}} (y^t - \tau^t(\mathbf{x}))^2 p(y^t|\mathbf{x}) p(\mathbf{x}, t) dy^t d\mathbf{x} dt$ and $A_{y^t}(p(\mathbf{x}, t)) = \int_{\mathcal{X} \times \{0,1\} \times \mathcal{Y}} |y^t - \tau^t(\mathbf{x})| p(y^t|\mathbf{x}) p(\mathbf{x}, t) dy^t d\mathbf{x} dt \; \forall t \in \{0, 1\}$.*
*Let loss function $L$ be the squared loss. Then we have:*

$$\epsilon_{PEHE}(h, \Phi) \leq 2(\epsilon_{CF}(h, \Phi) + \epsilon_F(h, \Phi) - 2\sigma_y^2). \tag{2}$$

*Let loss function $L$ be the absolute loss. Then we have:*

$$\epsilon_{PEHE}(h, \Phi) \leq \epsilon_{CF}(h, \Phi) + \epsilon_F(h, \Phi) + 2A_y. \tag{3}$$

Lemma 1 reveals that the ITE error $\epsilon_{PEHE}$ is closely connected with the factual error $\epsilon_F$ and counterfactual $\epsilon_{CF}$, as well as a constant $\sigma_y^2$ (or $A_y$) that is unrelated with functions $h$ and $\Phi$. Here, $\sigma_y^2$ is the smaller value of the variance in $Y^t$ w.r.t. the distribution $p(\mathbf{x}, t)$ and the variance in $Y^{1-t}$ w.r.t. $p(\mathbf{x}, 1 - t)$, and $A_y$ is the larger value of the absolute deviation in $Y^t$ w.r.t. the distribution $p(\mathbf{x}, t)$ and the absolute deviation in $Y^{1-t}$ w.r.t. the distribution $p(\mathbf{x}, 1 - t)$. The proof of Lemma 1 is deferred to Section A.2. Note that equation (2) corresponds to the result presented in Shalit et al. (2017), while equation (3) is our new result, which supplements the case when $L$ denotes the absolute loss.

### 4.1 Wasserstein Distance Guided Error Bounds

Previous causal learning models commonly adopt the Wasserstein distance guided approach to seek representation balancing. In this subsection, we first give the definition of Wasserstein distance (Cuturi & Doucet, 2014) by introducing the Integral Probability Metric (IPM) (Sriperumbudur et al., 2012) defined in Definition 2. Then we state the theorem regarding the upper bounds for counterfactual error $\epsilon_{CF}$ and ITE error $\epsilon_{PEHE}$ using Wasserstein distance in Theorem 1.

**Definition 2.** *Let $\mathcal{G}$ be a function family consisting of functions $g : \mathcal{S} \to \mathbb{R}$. For a pair of distributions $p_1$, $p_2$ over $\mathcal{S}$, the Integral Probability Metric is defined as*

$$IPM_{\mathcal{G}}(p_1, p_2) := \sup_{g \in \mathcal{G}} |\int_{\mathcal{S}} g(s)(p_1(s) - p_2(s))ds|.$$

If $\mathcal{G}$ is the family of 1-Lipschitz functions, we can obtain the so-called 1-Wasserstein distance, denoted by $Wass(p_1, p_2)$. Next, we present the bounds for counterfactual error $\epsilon_{CF}$ and ITE error $\epsilon_{PEHE}$ using Wasserstein distance in Theorem 1.

**Theorem 1.** *Let $\Phi : \mathcal{X} \to \mathcal{R}$ be an invertible representation with $\Psi$ being its inverse. Define $\sigma_y^2 = \min\{\sigma_{y^t}^2(p(\mathbf{x}, t)), \sigma_{y^t}^2(p(\mathbf{x}, 1 - t))\}$ and $A_y = \max\{A_{y^t}(p(\mathbf{x}, t)), A_{y^t}(p(\mathbf{x}, 1 - t))\}$ $\forall t \in \{0, 1\}$, where $\sigma_{y^t}^2(p(\mathbf{x}, t)) = \int_{\mathcal{X} \times \{0,1\} \times \mathcal{Y}} (y^t - \tau^t(\mathbf{x}))^2 p(y^t|\mathbf{x}) p(\mathbf{x}, t) dy^t d\mathbf{x} dt$ and $A_{y^t}(p(\mathbf{x}, t)) = \int_{\mathcal{X} \times \{0,1\} \times \mathcal{Y}} |y^t - \tau^t(\mathbf{x})| p(y^t|\mathbf{x}) p(\mathbf{x}, t) dy^t d\mathbf{x} dt$ $\forall t \in \{0, 1\}$. Let $p_{\Phi}^{T=1}(\mathbf{r})$, $p_{\Phi}^{T=0}(\mathbf{r})$ be as defined before, $h : \mathcal{R} \times \{0, 1\} \to \mathcal{Y}$, $u := Pr(T = 1)$ and $\mathcal{G}$ be the family of 1-Lipschitz functions. Assume there exists a constant $B_{\Phi} \geq 0$, such that for $t \in \{0, 1\}$, the function $g_{\Phi,h}(\mathbf{r}, t) := \frac{1}{B_{\Phi}} \cdot \ell_{h,\Phi}(\Psi(\mathbf{r}), t) \in \mathcal{G}$. Given a loss function $L$, we have*

$$\epsilon_{CF}(h, \Phi) \leq (1 - u) \cdot \epsilon_F^{T=1}(h, \Phi) + u \cdot \epsilon_F^{T=0}(h, \Phi) + B_{\Phi} \cdot Wass(p_{\Phi}^{T=1}, p_{\Phi}^{T=0}). \tag{4}$$

*Let loss function $L$ be the squared loss. Then we have:*

$$\epsilon_{PEHE}(h, \Phi) \leq 2(\epsilon_F^{T=1}(h, \Phi) + \epsilon_F^{T=0}(h, \Phi) + B_{\Phi} \cdot Wass(p_{\Phi}^{T=1}, p_{\Phi}^{T=0}) - 2\sigma_y^2). \tag{5}$$

*Let loss function $L$ be the absolute loss. Then we have:*

$$\epsilon_{PEHE}(h, \Phi) \leq \epsilon_F^{T=1}(h, \Phi) + \epsilon_F^{T=0}(h, \Phi) + B_{\Phi} \cdot Wass(p_{\Phi}^{T=1}, p_{\Phi}^{T=0}) + 2A_y. \tag{6}$$

Theorem 1 reveals that the ITE error is closely tied to the factual error $\epsilon_F$ and the Wasserstein distance between treated and controlled groups in the representation space. This theorem provides a theoretical foundation for representation balancing models based on group distance minimization (Section 5.1.1). The proof of Theorem 1 is deferred to Section A.3. Note that equation (5) corresponds to the result presented in Shalit et al. (2017), while equation (6) is our new result, which supplements the case when $L$ denotes the absolute loss.

### 4.2 $\mathcal{H}$-divergence Guided Error Bounds

In most representation balancing literature, the models mainly rely on Wasserstein distance guided error bounds as discussed in Section 4.1. In this subsection, we will focus on establishing $\mathcal{H}$-divergence guided error bounds for counterfactual and ITE estimations in representation balancing approach. We first give the definition of $\mathcal{H}$-divergence (Ben-David et al., 2006) in Definition 3. Then we state the theorem regarding the upper bounds for counterfactual error $\epsilon_{CF}$ and ITE error $\epsilon_{PEHE}$ using $\mathcal{H}$-divergence in Theorem 2.

**Definition 3.** *Given a pair of distributions $p_1$, $p_2$ over $\mathcal{S}$, and a hypothesis binary function class $\mathcal{H}$, the $\mathcal{H}$-divergence between $p_1$ and $p_2$ is defined as*

$$d_{\mathcal{H}}(p_1, p_2) := 2 \sup_{\eta \in \mathcal{H}} |Pr_{p_1}[\eta(s) = 1] - Pr_{p_2}[\eta(s) = 1]|. \tag{7}$$

**Theorem 2.** *Let $\Phi : \mathcal{X} \to \mathcal{R}$ be an invertible representation with $\Psi$ being its inverse. Define $\sigma_y^2 = \min\{\sigma_{y^t}^2(p(\mathbf{x}, t)), \sigma_{y^t}^2(p(\mathbf{x}, 1 - t))\}$ and $A_y = \max\{A_{y^t}(p(\mathbf{x}, t)), A_{y^t}(p(\mathbf{x}, 1 - t))\}$ $\forall t \in \{0, 1\}$,*

*where* $\sigma_{y^t}^2(p(\mathbf{x},t)) = \int_{\mathcal{X}\times\{0,1\}\times\mathcal{Y}}(y^t - \tau^t(\mathbf{x}))^2 p(y^t|\mathbf{x})p(\mathbf{x},t)dy^t d\mathbf{x}dt$ *and* $A_{y^t}(p(\mathbf{x},t)) = \int_{\mathcal{X}\times\{0,1\}\times\mathcal{Y}}|y^t - \tau^t(\mathbf{x})|p(y^t|\mathbf{x})p(\mathbf{x},t)dy^t d\mathbf{x}dt$ $\forall t \in \{0,1\}$. *Let* $p_\Phi^{T=1}(\mathbf{r})$, $p_\Phi^{T=0}(\mathbf{r})$ *be as defined before,* $h : \mathcal{R}\times\{0,1\} \to \mathcal{Y}$, $u := Pr(T=1)$ *and* $\mathcal{H}$ *be the family of binary functions. Assume that there exists a constant* $K \geq 0$ *such that* $\int_{\mathcal{Y}}L(y,y')dy \leq K$ $\forall y' \in \mathcal{Y}$. *Given a loss function L, we have*

$$\epsilon_{CF}(h,\Phi) \leq (1-u)\cdot\epsilon_F^{T=1}(h,\Phi) + u\cdot\epsilon_F^{T=0}(h,\Phi) + \frac{K}{2}d_\mathcal{H}(p_\Phi^{T=1}, p_\Phi^{T=0}). \tag{8}$$

*Let loss function L be the squared loss. Then we have:*

$$\epsilon_{PEHE}(h,\Phi) \leq 2(\epsilon_F^{T=1}(h,\Phi) + \epsilon_F^{T=0}(h,\Phi) + \frac{K}{2}d_\mathcal{H}(p_\Phi^{T=1}, p_\Phi^{T=0}) - 2\sigma_y^2). \tag{9}$$

*Let loss function L be the absolute loss. Then we have:*

$$\epsilon_{PEHE}(h,\Phi) \leq \epsilon_F^{T=1}(h,\Phi) + \epsilon_F^{T=0}(h,\Phi) + \frac{K}{2}d_\mathcal{H}(p_\Phi^{T=1}, p_\Phi^{T=0}) + 2A_y. \tag{10}$$

Theorem 2 reveals that the ITE error is closely connected with the factual error $\epsilon_F$ and the $\mathcal{H}$-divergence between treated and controlled samples in the representation space. This new theoretical result provides a theoretical foundation for representation balancing models based on individual propensity confusion (Section 5.1.2). The proof of Theorem 2 is deferred to Section A.4.

## 5 Method

In the preceding section, we have stated the theoretical foundations for representation balancing methods, which are the Wasserstein distance guided error bounds (results in Shalit et al. (2017)) and $\mathcal{H}$-divergence guided error bounds (Our results). Moving on to Section 5.1, we will begin by introducing representation balancing methods without decomposed patterns. Specifically, Section 5.1.1 revisits a Wasserstein distance based representation balancing network GNet, and Section 5.1.2 demonstrates how Theorem 2 can be connected with individual propensity confusion, helping us to build a $\mathcal{H}$-divergence based representation balancing network INet. Subsequently, in Section 5.2, we will introduce how to design a representation balancing method within the scheme of decomposed patterns, based on the PDIG and PPBR methods (Section 5.2.1). The final proposed model DIGNet is presented in Section 5.2.2.

### 5.1 Representation Balancing without Decomposed Patterns

In representation balancing models, given the input data tuples $(\mathbf{x},\mathbf{t},\mathbf{y}) = \{(\mathbf{x}_i,t_i,y_i)\}_{i=1}^N$, the original covariates $\mathbf{x}$ are extracted by some representation function $\Phi(\cdot)$, and representations $\Phi(\mathbf{x})$ are then fed into the outcome functions $h^1(\cdot) := h(\cdot,1)$ and $h^0(\cdot) := h(\cdot,0)$ that estimate the potential outcome $y^1$ and $y^0$, respectively. Finally, the factual outcome can be predicted by $h^t(\cdot) = th^1(\cdot) + (1-t)h^0(\cdot)$, and the corresponding outcome loss is

$$\mathcal{L}_y(\mathbf{x},\mathbf{t},\mathbf{y};\Phi,h^t) = \frac{1}{N}\sum_{i=1}^N L(h^t(\Phi(\mathbf{x}_i)),y_i). \tag{11}$$

The loss function $\mathcal{L}_y$ approximates the factual error $\epsilon_F$ appeared in Theorems 1 and 2. Minimizing $\mathcal{L}_y$ also corresponds to the Principle I as mentioned in the Introduction.

#### 5.1.1 GNet: Group Distance Minimization Guided Network

The ***group distance minimization*** focuses on learning representations that minimize the distance between the treated and controlled groups, and the corresponding theoretical foundation is supported by Wasserstein distance guided counterfactual and ITE error bounds (Theorem 1). Previous causal inference methods (e.g., Shalit et al. (2017); Yao et al. (2018); Zhang et al. (2020); Huang et al. (2022a)) commonly adopt Wasserstein

distance to achieve group distance minimization. Specifically, these methods aim to minimize the empirical approximation of $\mathcal{L}_G(\mathbf{x}, \mathbf{t}; \Phi) = Wass\left(\{\Phi(\mathbf{x}_i)\}_{i:t_i=0}, \{\Phi(\mathbf{x}_i)\}_{i:t_i=1}\right)$ to learn balancing patterns. If we denote $\Phi_E(\cdot)$ by the feature extractor that extracts the original covariates $\mathbf{x}$, then the objective function designed on Theorem 1 is

$$\min_{\Phi_E, h^t} \quad \mathcal{L}_y(\mathbf{x}, \mathbf{t}, \mathbf{y}; \Phi_E, h^t) + \alpha_1 \mathcal{L}_G(\mathbf{x}, \mathbf{t}; \Phi_E). \tag{12}$$

Since the objective is to learn balancing patterns by minimizing the distributional distance between treated and controlled groups, i.e., group distance minimization, we refer to a model with the objective in equation (12) as **GNet**. For the reader's convenience, we illustrate the structure of GNet in Figure 2(a). Note that CFRNet (Shalit et al., 2017) is also the category of GNet.

### 5.1.2 INet: Individual Propensity Confusion Guided Network

In the field of causal inference, the propensity score plays a central role because it characterizes the probability that one receives treatment (Rosenbaum & Rubin, 1983). For example, the propensity score has been widely employed in prior literature for matching (Caliendo & Kopeinig, 2008) or weighting (Austin & Stuart, 2015) purposes. In this paper, we emphasize that the propensity score also plays an important role in representation balancing, where it serves as a natural indicator of the adequacy of leanred balancing patterns. Specifically, we propose the concept of individual propensity confusion, which aims to learn representations that are difficult to utilize for characterizing the propensity of each individual being treated or controlled. The underlying theoretical foundation is upon the $\mathcal{H}$-divergence guided ITE error bounds derived in Theorem 2. Specifically, equations 9 and 10 in Theorem 2 highlight the significance of minimizing the generalization bound associated with factual outcome error and the $\mathcal{H}$-divergence between treated and controlled representations in reducing ITE errors. Subsequently, we will present the details of achieving representation balancing by reducing the $\mathcal{H}$-divergence between treated and controlled samples in the representation space.

Let $\mathbf{1}(a)$ be an indicator function that gives 1 if $a$ is true, and $\mathcal{H}$ be the family of binary functions as defined in Theorem 2. To achieve representation balancing, the objective function designed on Theorem 2 should aim to minimize the empirical $\mathcal{H}$-divergence $\hat{d}_{\mathcal{H}}(p_\Phi^{T=1}, p_\Phi^{T=0})$ such that

$$\hat{d}_{\mathcal{H}}(p_\Phi^{T=1}, p_\Phi^{T=0}) = 2\left(1 - \min_{\eta \in \mathcal{H}}\left[\frac{1}{N}\sum_{i:\eta(\Phi(\mathbf{x}_i))=0}\mathbf{1}[t_i = 1] + \frac{1}{N}\sum_{i:\eta(\Phi(\mathbf{x}_i))=1}\mathbf{1}[t_i = 0]\right]\right). \tag{13}$$

The "min" part in equation (13) indicates that the optimal classifier $\eta^* \in \mathcal{H}$ minimizes the classification error between the estimated treatment $\eta^*(\Phi(\mathbf{x}_i))$ and the observed treatment $t_i$, i.e., discriminating whether $\Phi(\mathbf{x}_i)$ is a control ($T=0$) or treatment ($T=1$). Equation (13) suggests that $\hat{d}_{\mathcal{H}}(p_\Phi^{T=1}, p_\Phi^{T=0})$ will be large if $\eta^*$ can easily distinguish whether $\Phi(\mathbf{x}_i)$ is treated or controlled. In contrast, $\hat{d}_{\mathcal{H}}(p_\Phi^{T=1}, p_\Phi^{T=0})$ will be small if it is hard for $\eta^*$ to determine whether $\Phi(\mathbf{x}_i)$ is treated or controlled. Therefore, the prerequisite of a small $\mathcal{H}$-divergence is to find a map $\Phi$ such that any classifier $\eta \in \mathcal{H}$ will get confused about the probability of $\Phi(\mathbf{x}_i)$ being treated or controlled. To achieve this goal, similar to the strategy of empirical approximation of $\mathcal{H}$-divergence (Ganin et al., 2016), we define a discriminator $\pi(\mathbf{r}) : \mathcal{R} \to [0, 1]$ that estimates the propensity score of $\mathbf{r}$, which can be regarded as a surrogate for $\eta(\mathbf{r})$. The classification error for the $i^{th}$ individual can be empirically approximated by the cross-entropy loss between $\pi(\Phi(\mathbf{x}_i))$ and $t_i$:

$$\mathcal{L}_t(t_i, \pi(\Phi(\mathbf{x}_i))) = -\left[t_i \log \pi(\Phi(\mathbf{x}_i)) + (1 - t_i)\log(1 - \pi(\Phi(\mathbf{x}_i)))\right]. \tag{14}$$

As a consequence, we aim to find an optimal discriminator $\pi^*$ for equation (13) such that $\pi^*$ maximizes the probability that treatment is correctly classified:

$$\max_{\pi \in \mathcal{H}} \mathcal{L}_I(\mathbf{x}, \mathbf{t}; \Phi, \pi) = \max_{\pi \in \mathcal{H}}\left[-\frac{1}{N}\sum_{i=1}^{N}\mathcal{L}_t(t_i, \pi(\Phi(\mathbf{x}_i)))\right]. \tag{15}$$

Given the feature extractor $\Phi_E(\cdot)$, the objective of INet can be formulated as a min-max game:

$$\min_{\Phi_E, h^t} \max_{\pi} \quad \mathcal{L}_y(\mathbf{x}, \mathbf{t}, \mathbf{y}; \Phi_E, h^t) + \alpha_2 \mathcal{L}_I(\mathbf{x}, \mathbf{t}; \Phi_E, \pi). \tag{16}$$

In the maximization, the discriminator $\pi$ is trained to maximize the probability that treatment is correctly classified. This forces $\pi(\Phi_E(\mathbf{x}_i))$ closer to the true propensity score $e(\mathbf{x}_i)$. In the minimization, the feature extractor $\Phi_E$ is trained to fool the discriminator $\pi$. This confuses $\pi$ such that $\pi(\Phi_E(\mathbf{x}_i))$ cannot correctly specify the true propensity score $e(\mathbf{x}_i)$. Eventually, the representations are balanced as the adversarial process makes it difficult for $\pi$ to determine the propensity of each individual being treated or controlled. We refer to this process as ***individual propensity confusion***. Such an adversarial learning technique has been widely used in domain adaptation (e.g., Ganin et al. (2016); Tzeng et al. (2017)) and fair representation learning (e.g., Edwards & Storkey (2015); Madras et al. (2018)) to learn domain-invariant and fair representations. For the reader's convenience, we illustrate the structure of INet in Figure 2(b).

## 5.2 Representation Balancing with Decomposed Patterns

### 5.2.1 The Proposed PDIG and PPBR Methods

**PDIG.** Although Theorems 1 and 2 provide solid theoretical foundation for GNet (model proposed by Shalit et al. (2017)) and INet (model proposed by us), both of these model types still encounter the inherent trade-off between representation balancing and outcome modeling. To this end, we expect to capture more effective balancing patterns by learning **P**atterns **D**ecomposed with **I**ndividual propensity confusion and **G**roup distance minimization **(PDIG)**. More specifically, the covariates $\mathbf{x}$ are extracted by the feature extractor $\Phi_E(\cdot)$, and then $\Phi_E(\mathbf{x})$ are fed into two distinct balancing networks $\Phi_G(\cdot)$ and $\Phi_I(\cdot)$ for group distance minimization and individual propensity confusion, respectively. In summary, PDIG decomposes the balancing patterns into two distinct parts, group distance minimization and individual propensity confusion, which are respectively achieved by the following loss functions:

$$\min_{\Phi_G} \quad \mathcal{L}_G(\mathbf{x}, \mathbf{t}; \Phi_G \circ \Phi_E) \tag{17}$$

$$\min_{\Phi_I} \max_{\pi} \quad \mathcal{L}_I(\mathbf{x}, \mathbf{t}; \Phi_I \circ \Phi_E, \pi). \tag{18}$$

Here, $\circ$ denotes the composition of two functions, indicating that $\Phi(\cdot)$ in $\mathcal{L}_G(\mathbf{x}, \mathbf{t}; \Phi)$ and $\mathcal{L}_I(\mathbf{x}, \mathbf{t}; \Phi, \pi)$ are replaced by $\Phi_G(\Phi_E(\cdot))$ and $\Phi_I(\Phi_E(\cdot))$, respectively.

**PPBR.** Motivated by the discussion in Section 1, we design a framework that is capable of capturing **P**atterns of **P**re-balancing and **B**alancing **R**epresentations **(PPBR)** to mitigate potential over-balancing issue mentioned in the Introduction, aiming to preserve information that is useful for outcome predictions. In the PPBR method, the balancing patterns $\Phi_G(\Phi_E(\mathbf{x}))$ and $\Phi_I(\Phi_E(\mathbf{x}))$ are first learned over $\Phi_G$ and $\Phi_I$, while $\Phi_E$ is remained fixed as pre-balancing patterns. Furthermore, we concatenate the balancing patterns $\Phi_G(\Phi_E(\mathbf{x}))$ and $\Phi_I(\Phi_E(\mathbf{x}))$ with the pre-balancing representations $\Phi_E(\mathbf{x})$ as attributes for outcome prediction. As a result, the proxy features used for outcome predictions are $\Phi_E(\mathbf{x}) \oplus \Phi_G(\Phi_E(\mathbf{x})) \oplus \Phi_I(\Phi_E(\mathbf{x}))$, where $\oplus$ indicates the concatenation by column. For example, if $\mathbf{a} = [1, 2]$ and $\mathbf{b} = [3, 4]$, then $\mathbf{a} \oplus \mathbf{b} = [1, 2, 3, 4]$. Consequently, representation balancing is accomplished over $\Phi_G$ and $\Phi_I$, rather than $\Phi_E$. Even if there may be a loss of information relevant to outcome prediction in $\Phi_G$ and $\Phi_I$, the pre-balancing patterns $\Phi_E$ can still effectively preserve such information. Finally, the objective function with regard to outcome modeling under PPBR method becomes

$$\min_{\Phi_E, \Phi_I, \Phi_G, h^t} \quad \mathcal{L}_y(\mathbf{x}, \mathbf{t}, \mathbf{y}; \Phi_E \oplus (\Phi_I \circ \Phi_E) \oplus (\Phi_G \circ \Phi_E), h^t). \tag{19}$$

### 5.2.2 The Proposed DIGNet

Combining with PDIG and PPBR, we propose a new neural **Net**work model that incorporates **D**ecomposed patterns with **I**ndividual propensity confusion and **G**roup distance minimization, which we call **DIGNet**.

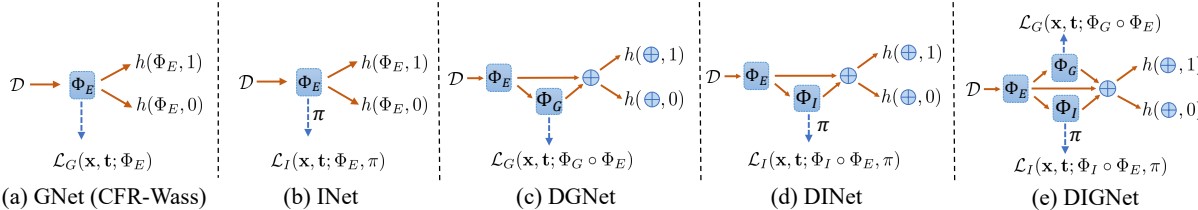

Figure 2: Illustrations of the network architecture of the five models studied in Section 6.

The objective of DIGNet is separated into four stages:

$$\min_{\Phi_G} \quad \alpha_1 \mathcal{L}_G(\mathbf{x}, \mathbf{t}; \Phi_G \circ \Phi_E), \tag{20}$$

$$\max_{\pi} \quad \alpha_2 \mathcal{L}_I(\mathbf{x}, \mathbf{t}; \Phi_I \circ \Phi_E, \pi), \tag{21}$$

$$\min_{\Phi_I} \quad \alpha_2 \mathcal{L}_I(\mathbf{x}, \mathbf{t}; \Phi_I \circ \Phi_E, \pi), \tag{22}$$

$$\min_{\Phi_E, \Phi_I, \Phi_G, h^t} \quad \mathcal{L}_y(\mathbf{x}, \mathbf{t}, \mathbf{y}; \Phi_E \oplus (\Phi_I \circ \Phi_E) \oplus (\Phi_G \circ \Phi_E), h^t). \tag{23}$$

Within each iteration, DIGNet minimizes the group distance through equation 20, and plays an adversarial game to achieve propensity confusion through equation 21 and equation 22. In equation 23, DIGNet updates both the pre-balancing patterns $\Phi_E$ and balancing patterns $\Phi_I, \Phi_G$, along with the outcome function $h^t$ to minimize the outcome prediction loss. For the reader's convenience, we illustrate the structure of DIGNet in Figure 2(e).

## 5.3 Insights of Representation Balancing with Decomposed Patterns

Our proposed DIGNet model builds upon the PDIG and PPBR methods. The PPBR method is relatively straightforward, as it forms more flexible predictor $(\Phi_E \oplus (\Phi_I \circ \Phi_E))$ (or $(\Phi_E \oplus (\Phi_G \circ \Phi_E)))$ compared to the solely predictor $\Phi_E$. Therefore, incorporating both pre-balancing and balancing patterns is helpful in enhancing the model's complexity and its ability to capture more useful information for outcome prediction. However, there still remains further exploration to better understand why the PDIG method is effective. The DIGNet model aims to learn balancing patterns based on both Wasserstein distance and $\mathcal{H}$-divergence. At first glance, one might assume that incorporating both distances could be redundant, as one distance seems naturally to imply the other. In this section, we gain some insights of these two divergence metrics. First, we provide a systematic discussion on the properties of Wasserstein distance and $\mathcal{H}$-divergence. In addition, we utilize a toy example to illustrate their distinct abilities in capturing distributional disparity. Further, we use this example to aid readers in better understanding the trade-off problem encountered in representation balancing models (Figure 5). Finally, we establish a connection between our method and the Elastic Net method and Multi-task learning approach, which offers valuable insights and explanations regarding the intuition behind involving both metrics as regularizations.

### 5.3.1 Properties of Wasserstein Distance and $\mathcal{H}$-Divergence

Wasserstein distance and $\mathcal{H}$-Divergence possess distinct theoretical properties. The effectiveness of the Wasserstein distance in measuring distributional differences for classification tasks in domain adaptation has been demonstrated in Shen et al. (2018). Furthermore, Shalit et al. (2017) highlights the potential of Wasserstein distance in representation balancing models for ITE estimation, which significantly outperforms traditional ITE estimation methods. Wasserstein distance is also widely adopted in other research domains, such as fair representation learning (Jiang et al., 2020), as discussed in Section 2. Its prevalence stems from its strong capability to capture better diversities compared to $\mathcal{H}$-Divergence (Shui et al., 2020). Studies have proven that under certain conditions, it is possible to bound $\mathcal{H}$-Divergence using Wasserstein distance (Villani et al., 2009; Shui et al., 2020), which provides a reasonable explanation for the overall superiority of

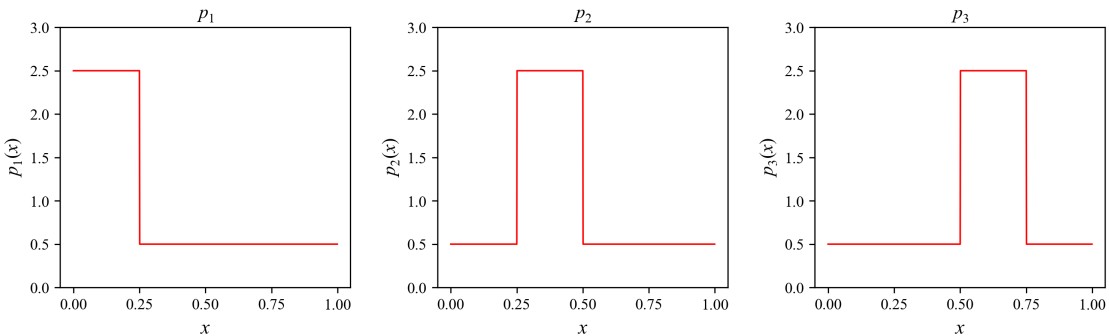

Figure 3: Distributions of $p_1(x)$, $p_2(x)$, and $p_3(x)$ in the example of Section 5.3.1.

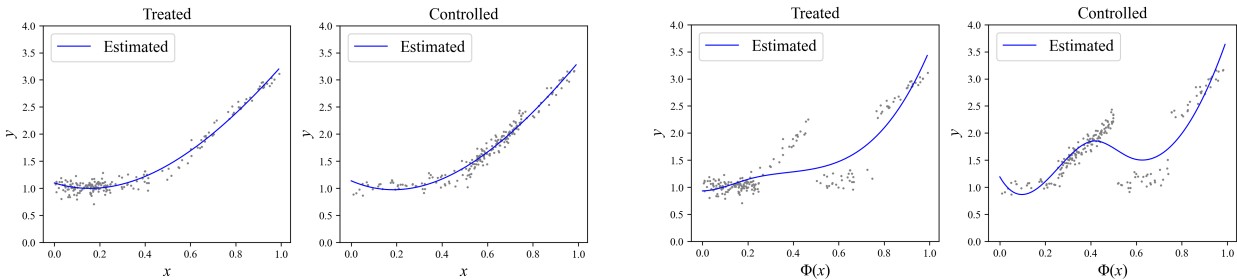

(a) Model fitting using $(X, Y)$. The probability density functions of $X^{\text{treat}}$ and $X^{\text{control}}$ are $p_1$ and $p_3$, respectively.

(b) Model fitting using $(\Phi(X), Y)$. The probability density functions of $\Phi^{\text{treat}}(X)$ and $\Phi^{\text{control}}(X)$ are $p_1$ and $p_2$, respectively.

Figure 4: Model fitting using $(X, Y)$ and $(\Phi(X), Y)$ based on the example in Section 5.3.1.

the Wasserstein distance in learning domain-invariant features (Zhiri et al., 2022). However, it is important to note that this bound does not hold in general (Chae & Walker, 2020), suggesting that a smaller $\mathcal{H}$-divergence does not necessarily imply a smaller Wasserstein distance. To better illustrate the difference between these two measures, we provide a concrete example below.

**Toy example.** Consider the following three probability density functions $p_1(x)$, $p_2(x)$, and $p_3(x)$ defined over $x \in [0, 1]$:

$$p_1(x) = \begin{cases} 2.5, & \text{if } 0 \le x < 0.25 \\ 0.5, & \text{if } 0.25 \le x < 0.5 \\ 0.5, & \text{if } 0.5 \le x < 0.75 \\ 0.5, & \text{if } 0.75 \le x \le 1 \end{cases} \quad p_2(x) = \begin{cases} 0.5, & \text{if } 0 \le x < 0.25 \\ 2.5, & \text{if } 0.25 \le x < 0.5 \\ 0.5, & \text{if } 0.5 \le x < 0.75 \\ 0.5, & \text{if } 0.75 \le x \le 1 \end{cases} \quad p_3(x) = \begin{cases} 0.5, & \text{if } 0 \le x < 0.25 \\ 0.5, & \text{if } 0.25 \le x < 0.5 \\ 2.5, & \text{if } 0.5 \le x < 0.75 \\ 0.5, & \text{if } 0.75 \le x \le 1 \end{cases}.$$

The above three distributions are depicted in Figure 3. Further, we set the classifier in $\mathcal{H}$-divergence as $\eta(x) = \mathbf{1}\{x \ge p\}$, and set the order in Wasserstein distance as $p = 1$. By utilizing the definitions of $\mathcal{H}$-divergence and 1-Wasserstein distance, one can make a direct comparison between the discrepancy in $(p_1, p_2)$ and the discrepancy in $(p_1, p_3)$:

$$\begin{aligned} d_{\mathcal{H}}(p_1, p_2) &= d_{\mathcal{H}}(p_1, p_3); \\ Wass(p_1, p_2) &< Wass(p_1, p_3). \end{aligned} \tag{24}$$

Equation 24 confirms that Wasserstein distance is able to capture more diverse distributional disparities compared to $\mathcal{H}$-divergence. However, in the subsequent content, we will demonstrate that such an advantage might be a limitation in causal representation learning due to the trade-off problem.

**Understanding the trade-off.** The above example serves as simple evidence that supports the conclusion that Wasserstein distance can capture better diversities between distributions compared to $\mathcal{H}$-divergence (Shui et al., 2020). However, as discussed in Section 1, it is important to note that achieving a more balanced distribution does not necessarily ensure favorable generalization to counterfactuals. This is because the pursuit of balanced representations may inadvertently lead to a loss of information useful for factual outcome estimates. We will now use the above example to gain further understanding on this matter.

Consider a simple data-generating process where $X^{\text{treat}}$, the covariate in the treated group, follows the distribution $p_1(x)$, and $X^{\text{control}}$, the covariate in the controlled group, follows the distribution $p_3(x)$. Let the potential outcomes are generated by $Y^1 = \tau_1(X) + \epsilon_1$ and $Y^0 = \tau_0(X) + \epsilon_0$, where $\epsilon_1 \sim \mathcal{N}(0, 0.1)$ and $\epsilon_0 \sim \mathcal{N}(0, 0.1)$. Let the true potential outcome functions $\tau^1(x)$ and $\tau^0(x)$ be as follows:

$$\tau^1(x) = \tau^0(x) = (x^2 + 1)\mathbf{1}\{0 \leq x < 0.5\} + (4x - 0.75)\mathbf{1}\{0.5 \leq x \leq 1\}. \tag{25}$$

In addition, consider a representation function $\Phi(x)$ such that

$$\Phi(x) = x\mathbf{1}\{0 \leq x < 0.25\} + (x + 0.25)\mathbf{1}\{0.25 \leq x < 0.5\} + (x - 0.25)\mathbf{1}\{0.5 \leq x < 0.75\} + x\mathbf{1}\{0.75 \leq x \leq 1\}. \tag{26}$$

We can find $\Phi$ achieves representation balancing under Wasserstein distance measure, but does not under $\mathcal{H}$-divergence measure. In original data, $x^{\text{treat}}$ follows $p_1$ and $x^{\text{control}}$ follows $p_3$. After mapping $x$ to $\Phi(x)$, $\Phi^{\text{treat}}(x)$ follows $p_1$ and $\Phi^{\text{control}}(x)$ follows $p_2$. Consequently, based on the results in equation 24, we have

$$d_{\mathcal{H}}(p_\Phi^{\text{treat}}, p_\Phi^{\text{control}}) = d_{\mathcal{H}}(p_X^{\text{treat}}, p_X^{\text{control}});$$
$$Wass(p_\Phi^{\text{treat}}, p_\Phi^{\text{control}}) < Wass(p_X^{\text{treat}}, p_X^{\text{control}}). \tag{27}$$

We now investigate the fitting performance of models using $(x, y)$ and $(\Phi(x), y)$ to check whether there is a loss of outcome-related information during representation balancing. In Figure 4a and Figure 4b, we present scatter plots of samples from $(x, y)$ and $(\Phi(x), y)$ respectively, depicted as gray points. Following the approach of Kennedy (2023), we employ smoothing spline functions to fit these samples, and the estimated functions are illustrated in blue.

In Figure 4a, we observe that both $\tau_1$ and $\tau_0$ are well fitted using $(x, y)$, with their estimates being very close to each other. This is consistent with the setup of $\tau_1 = \tau_0$. In contrast, Figure 4b reveals that the fittings of $\tau_1$ and $\tau_0$ are inadequate using $(\Phi(x), y)$, resulting in substantially different estimates. The result of different estimates violates the setup of $\tau_1 = \tau_0$. In this case, a model based on Wasserstein distance would retain $\Phi$ due to its achievement of representation balancing. Unfortunately, $\Phi$ suffers from a loss of valuable information that is crucial for outcome prediction. In contrast, a model based on $\mathcal{H}$-divergence would not keep $\Phi$ since it does not contribute to reducing the domain distance compared to the original data. Fortunately, the original data preserve the information necessary for outcome modeling. Therefore, this example not only emphasizes the significance of incorporating both metrics but also highlights the importance of considering both pre-balancing patterns and balancing patterns.

### 5.3.2 Connection with other machine learning methods

In the previous sections, we have discussed the trade-off between factual outcome prediction and representation balancing in classic representation learning models. As part of our proposed improvements, DIGNet involves learning two distinct representations using Wasserstein distance and $\mathcal{H}$-divergence separately and concatenates the learned representations for outcome modeling. In this section, we will explore more detailed connections between our design and other machine learning methods.

**Connection with Elastic Net: balancing on two discrepancies.** Our DIGNet model involves two discrepancy metrics: Wasserstein distance and $\mathcal{H}$-divergence. We will now provide additional explanations on its connection with the Elastic Net method. In supervised learning, a regularization term is often incorporated during model training to mitigate the bias-variance trade-off. In the case of linear regression, Lasso (Tibshirani, 1996) and Ridge (Hoerl & Kennard, 1970) are proposed to improve the Ordinary Least Squares

(OLS) method, with Lasso involving $l_1$ regularization while Ridge involving $l_2$ regularization:

$$\text{Lasso:} \quad \min_{\boldsymbol{\beta} \in \mathcal{R}^d} \frac{1}{N} ||\mathbf{y} - \mathbf{X}\boldsymbol{\beta}||_2^2 + \alpha ||\beta||_1 = \min_{\boldsymbol{\beta} \in \mathcal{R}^d} \frac{1}{N} \sum_{i=1}^{N} (\mathbf{x}_i'\boldsymbol{\beta} - y_i)^2 + \alpha \sum_{j=1}^{d} |\beta_j|.$$

$$\text{Ridge:} \quad \min_{\boldsymbol{\beta} \in \mathcal{R}^d} \frac{1}{N} ||\mathbf{y} - \mathbf{X}\boldsymbol{\beta}||_2^2 + \alpha ||\beta||_2^2 = \min_{\boldsymbol{\beta} \in \mathcal{R}^d} \frac{1}{N} \sum_{i=1}^{N} (\mathbf{x}_i'\boldsymbol{\beta} - y_i)^2 + \alpha \sum_{j=1}^{d} \beta_j^2.$$

The different properties between $l_1$ regularization and $l_2$ regularization lead to distinct advantages and disadvantages between Lasso method and Ridge method. Given their differences, a method of Elastic Net (Zou & Hastie, 2005) is proposed by combing both $l_1$ regularization and $l_2$ regularization:

$$\text{Elastic Net:} \quad \min_{\boldsymbol{\beta} \in \mathcal{R}^d} \frac{1}{N} ||\mathbf{y} - \mathbf{X}\boldsymbol{\beta}||_2^2 + \alpha_1 ||\beta||_1 + \alpha_2 ||\beta||_2^2 = \min_{\boldsymbol{\beta} \in \mathcal{R}^d} \frac{1}{N} \sum_{i=1}^{N} (\mathbf{x}_i'\boldsymbol{\beta} - y_i)^2 + \alpha_1 \sum_{j=1}^{d} |\beta_j| + \alpha_2 \sum_{j=1}^{d} \beta_j^2.$$

The Elastic Net method integrates the strengths of two distinct approaches: the $l_1$ regularization term enforces sparsity, while the $l_2$ regularization maintains the grouping effect (Zhou, 2013; Narisetty, 2020). The Elastic Net has also motivated some research studies to adopt the idea of combining $l_1$ and $l_2$ regularizations in of deep neural networks (DNNs) (Kang et al., 2017; Chen et al., 2018; Hu et al., 2023; Xu et al., 2023a). Notably, a recent study (Xu et al., 2023a) presents an excess risk bound for Elastic Net Regularized DNNs. This finding provides supporting evidence that incorporating both $l_1$ and $l_2$ regularizations in a DNN model is reasonable. The insights gained from (Xu et al., 2023a) shed light on the theoretical explanation of our method, and even pave the way for exploring the integration of different divergence metrics in other research areas, such as domain adaptation, transfer learning, and fair representation learning.

**Connection with multi-task learning: balancing on two representations.** Our DIGNet model performs representation balancing on two distinct representations using Wasserstein distance and $\mathcal{H}$-divergence separately, and the learned representations are then concatenated for outcome modeling. We will now provide additional explanations regarding its connection with the multi-task learning method. In multi-task learning, distinct representations are learned for different tasks, with each task involving its own objective function. An important step in multi-task learning is integrating the information from these separately learned representations into a unified representation. One common approach is to concatenate the task-specific representations to form a joint representation, which effectively preserves the information from each task for outcome modeling (e.g.,Li et al. (2018); Baltrušaitis et al. (2018); Crawshaw (2020); Yan et al. (2021); Xu et al. (2023b)). For example, in an E-commerce application Liu et al. (2023), diverse types of user footprints are encoded using different representations with diverse objectives. The learned representations are then concatenated to make the final target prediction. Similarly, in another application Wu et al. (2018), user and product attentions are separately learned on two distinct representations, which are later concatenated for the final outcome prediction. In tasks such as image and text classification Hao et al. (2023), concatenating multiple representations has shown effective improvements in the classification performance, which is brought by the complementary information of each representation. Furthermore, a recent study on multi-view learning (Li et al., 2024) has also demonstrated that concatenating both the non-attention and attention representations of each view can prevent information loss in the final classification task.

**Summary of strengths and limitations.** Our method combines Wasserstein distance and $\mathcal{H}$-divergence for representation balancing to capture different types of balancing patterns compared to classic representation balancing models. This shares a similar intuition with the Elastic Net, which combines $l_1$ and $l_2$ regularizations to learn features with different properties. Notably, the two regularizations in Elastic Net are learned on a single parameter space with one objective, this provides more interpretability but might introduce a new trade-off. Different from Elastic Net, our DIGNet model concatenates the two distinct representations that are learned from two different tasks: Wasserstein distance guided and $\mathcal{H}$-divergence guided representation balancing. This aligns with the principle of multi-task learning. The concatenation fusion technique is extensively employed in numerous multi-task learning studies (Baltrušaitis et al., 2018), as it effectively preserves and integrates information from different tasks, leading to improved performance

Figure 5: T-SNE visualizations of the covariates as $\gamma$ varies. Red represents the treatment group and blue represents the control group. A larger $\gamma$ indicates a greater imbalance between the two groups.

in the final prediction objective (Hao et al., 2023; Li et al., 2024). However, it is crucial to acknowledge that this straightforward concatenation approach can present challenges when interpreting the specific role of each representation and can also increase model complexity (Jia et al., 2020).

## 6 Experiments

In non-randomized observational data, the ground truth regarding treatment effects remains inaccessible due to the absence of counterfactual information. Therefore, we use simulated data and semi-synthetic benchmark data to test the performance of our methods and other baseline models. In this section, we primarily investigate the three following questions:

**Q1.** Is PDIG helpful in ITE estimation through Path I in the Introduction, i.e., learning more effective balancing patterns without affecting factual outcome prediction?

**Q2.** Is PPBR helpful in ITE estimation through Path II in the Introduction, i.e., improving factual outcome prediction without affecting learning balancing patterns?

**Q3.** Can the proposed DIGNet model outperform other baseline models on benchmark dataset?

**Ablation models.** To investigate Q1 and Q2, we conducted ablation studies and designed two ablation models, **DGNet** and **DINet**, where DGNet (or DINet) can be considered as DIGNet without PDIG, and GNet (or INet) can be considered as DGNet (or DINet) without PPBR. The structures of DGNet and DINet are shown in Figure 2(c) and Figure 2(d), and the objectives of DGNet and DINet are deferred to Section A.6.

### 6.1 Experimental Settings

**Simulation data.** Previous causal inference works assess the model effectiveness by varying the distribution imbalance of covariates in treated and controlled groups at different levels (Yao et al., 2018; Yoon et al., 2018; Du et al., 2021). As suggested by Assaad et al. (2021), we draw 1000 observational data points from the following data generating strategy:

$$\mathbf{X}_i \sim \mathcal{N}(\mathbf{0}, \sigma^2 \cdot [\rho \mathbf{1}_p \mathbf{1}_p^{'} + (1 - \rho)\mathbf{I}_p]),$$
$$T_i \mid \mathbf{X}_i \sim \text{Bernoulli}(1/(1 + \exp(-\gamma \mathbf{X}_i))),$$
$$Y_i^0 = \boldsymbol{\beta}_\mathbf{0}' \mathbf{X}_i + \xi_i, \quad Y_i^1 = \boldsymbol{\beta}_\mathbf{1}' \mathbf{X}_i + \xi_i, \quad \xi_i \sim \mathcal{N}(0, 1).$$

Here, $\mathbf{1}_p$ denotes the $p$-dimensional all-ones vector and $\mathbf{I}_p$ denotes the identity matrix of size $p$. We fix $p = 10, \rho = 0.3, \sigma^2 = 2, \boldsymbol{\beta}_\mathbf{0}' = [0.3, ..., 0.3], \boldsymbol{\beta}_\mathbf{1}' = [1.3, ..., 1.3]$ and vary $\gamma \in \{0.25, 0.5, 0.75, 1, 1.5, 2, 3\}$ to yield different levels of selection bias. As seen in Figure 5, selection bias becomes more severe with $\gamma$ increasing. For each $\gamma$, we repeat the above data generating process to generate 30 different datasets, with each dataset split by the ratio of 56%/24%/20% as training/validation/test sets.

**Semi-synthetic data.** The IHDP dataset, introduced by Hill (2011), originates from the Infant Health and Development Program (IHDP). This program conducted a randomized controlled experiment in 1985

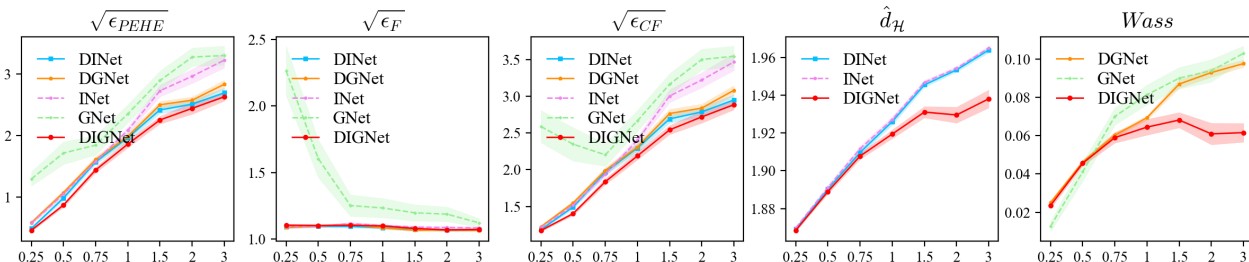

Figure 6: Plots of model performances on test set for different metrics as $\gamma$ varies in $\{0.25, 0.5, 0.75, 1, 1.5, 2, 3\}$. Each graph shows the average of 30 runs with standard errors shaded. Lower lines indicate lower values of the metric.

to investigate whether there is a positive causal effect of frequent high-quality child care and home visits (treatment) on cognitive scores (outcome). The collected data comprise 25-dimensional pre-treatment covariates, including measurements on the infants (e.g., birth weight, gender, head circumference), as well as measurements on the mothers during pregnancy (e.g., age, marital status, education, smoking and drinking habits). In order to create an observational dataset that involves selection bias, Hill excluded a subpopulation (children with nonwhite mothers) from the treated group. Consequently, the IHDP dataset exhibits a covariate shift, resulting in imbalanced treated and controlled groups. The final IHDP dataset consists of 747 samples, comprising 139 treated samples and 608 controlled samples. The potential outcomes were generated using setting A in the NPCI package Dorie (2021). We use the same 1000 datasets as used in Shalit et al. (2017), with each dataset split by the ratio of 63%/27%/10% as training/validation/test sets.

**Models and metrics.** In simulation experiments, we perform comprehensive comparisons between INet, GNet, DINet, DGNet, and DIGNet in terms of the mean and standard error for the following metrics: $\sqrt{\epsilon_{PEHE}}$, $\sqrt{\epsilon_{CF}}$, and $\sqrt{\epsilon_F}$ with $L$ defined in Definition 1 being the squared loss, as well as the empirical approximations of $Wass(p_\Phi^{T=1}, p_\Phi^{T=0})$ and $d_\mathcal{H}(p_\Phi^{T=1}, p_\Phi^{T=0})$ (denoted by $Wass$ and $\hat{d}_\mathcal{H}$, respectively). Note that as shown in Figure 2, $Wass$ is over $\Phi_E$ for GNet while over $\Phi_G$ for DGNet and DIGNet; $\hat{d}_\mathcal{H}$ is over $\Phi_E$ for INet while over $\Phi_I$ for DINet and DIGNet. To analyze the source of gain and ensure fair comparison in simulation studies, we fix hyperparameters across all models. This way is consistent with Curth & van der Schaar (2021). We apply an early stopping rule to all models as Shalit et al. (2017) do. In IHDP experiment, we use $\sqrt{\epsilon_{PEHE}}$, as well as an additional metric $\epsilon_{ATE} = |\hat{\tau}_{ATE} - \tau_{ATE}|$ to evaluate performances of various causal models (see them in Table 6). More descriptions of the implementation details, as well as the analysis of training time, training stability, and hyperparameter sensitivity, are deferred to Section A.5.

**Device.** All the experiments are run on Dell 7920 with one 16-core Intel Xeon Gold 6250 3.90GHz CPU and three NVIDIA Quadro RTX 6000 GPUs.

## 6.2 Results and Analysis

### 6.2.1 Preliminary Experimental Results

In this part, we first make a general comparison between different models with the degree of covariate imbalance increasing, and the relevant results are shown in Figure 6. There are four main observations: (1) DIGNet attains the lowest $\sqrt{\epsilon_{PEHE}}$ across all datasets, while GNet have inferior performances than other models; (2) DINet and DGNet outperform INet and GNet regarding $\sqrt{\epsilon_{CF}}$ and $\sqrt{\epsilon_{PEHE}}$; (3) INet, DINet, and DGNet have comparable performance to DIGNet in terms of factual outcome estimations ($\sqrt{\epsilon_F}$), but cannot compete with DIGNet in terms of counterfactual estimations ($\sqrt{\epsilon_{CF}}$) or ITE estimations ($\sqrt{\epsilon_{PEHE}}$); (4) DIGNet achieves smaller $\hat{d}_\mathcal{H}$ (or $Wass$) than DINet and INet (or DGNet and GNet), especially when the covariate shift problem is severe (e.g., when $\gamma > 1$). In conclusion, the above study has produced several noteworthy findings. Firstly, finding (1) reveals that our proposed DIGNet model consistently performs well in ITE estimation. Secondly, as indicated by finding (2), implementing the PPBR approach can enhance the

Table 1: Ablation study for PDIG: Mean $\pm$ std of each metric averaged across 30 runs on test set when $\gamma = 3$. Lower value is better.

| | $\sqrt{\epsilon_F}$ | $\sqrt{\epsilon_{CF}}$ | $\hat{d}_{\mathcal{H}}$ | $Wass$ |
|---|---|---|---|---|
| DIGNet | $1.07 \pm 0.01$ | $2.89 \pm 0.07$ | $1.94 \pm 0.00$ | $0.06 \pm 0.00$ |
| DINet | $1.07 \pm 0.01$ | $2.95 \pm 0.07$ | $1.94 \pm 0.00$ | - |
| DGNet | $1.07 \pm 0.01$ | $3.08 \pm 0.07$ | - | $0.10 \pm 0.00$ |

Table 2: Ablation study for PPBR: Mean $\pm$ std of each metric averaged across 30 runs on test set when $\gamma = 3$. Lower value is better.

| | $\sqrt{\epsilon_F}$ | $\sqrt{\epsilon_{CF}}$ | $\hat{d}_{\mathcal{H}}$ | $Wass$ |
|---|---|---|---|---|
| DGNet | $1.07 \pm 0.01$ | $3.08 \pm 0.07$ | - | $0.10 \pm 0.00$ |
| GNet | $1.12 \pm 0.03$ | $3.55 \pm 0.14$ | - | $0.10 \pm 0.00$ |
| DINet | $1.07 \pm 0.01$ | $2.95 \pm 0.07$ | $1.96 \pm 0.00$ | - |
| INet | $1.08 \pm 0.01$ | $3.47 \pm 0.12$ | $1.96 \pm 0.00$ | - |

predictive accuracy of factual and counterfactual outcomes. Lastly, findings (3) and (4) highlight the role of PDIG structure in enhancing the simultaneous reinforcement and complementarity of group distance minimization and individual propensity confusion, resulting in more balanced representations. Our subsequent analysis will step beyond these preliminary conclusions to gain a deeper understanding of the effectiveness of the proposed methods.

### 6.2.2 Further Ablation Studies

So far our preliminary observations have show that the relationship between the ITE errors of each model is: DIGNet<DINet<INet and DIGNet<DGNet<GNet. To further explore how PDIG and PPBR contribute to the improvement of ITE estimations, we choose the case with high selection bias ($\gamma = 3$) to analyze the source of gain for PDIG and PPBR. We report model performances (mean $\pm$ std) of each specific metric averaged across 30 runs on test set in Table 1 and Table 2. We also report model performances (mean $\pm$ std) averaged over 30 training and test sets in Table 3. Below we discuss the source of gain in detail.

**Ablation study for PDIG.** *The PDIG structure is manifest to be effective in capturing more effective balancing patterns, without affecting factual outcome predictions.* As depicted in Figure 6, DIGNet exhibits more balanced representations, irrespective of whether the discrepancy is measured by $\hat{d}_{\mathcal{H}}$ or $Wass$, while DIGNet, DINet, and DGNet demonstrate comparable estimates of factual outcomes ($\sqrt{\epsilon_F}$). Two additional pieces of specific evidence can be observed from Table 1: (1) Despite the absence of PDIG in DINet and DGNet when compared to DIGNet, these three models exhibit very similar performance regarding $\sqrt{\epsilon_F}$, with the performance being $1.07 \pm 0.01$. This indicates that PDIG does not impact the factual estimation. (2) DIGNet achieves smaller $\hat{d}_{\mathcal{H}}$ with a $|1.94/1.96 - 1| = 1.0\%$ reduction (or $Wass$ with a $|0.06/0.10 - 1| = 40\%$ reduction) compared with DINet (or DGNet). This indicates that PDIG enables the model to learn more effective balancing patterns. The above two points indicate that PDIG can capture more effective balancing patterns, without affecting factual outcome predictions. This advantage translates into superior counterfactual estimation, with DIGNet reducing $\sqrt{\epsilon_{CF}}$ by $|2.89/2.95 - 1| = 2.0\%$ and $|2.89/3.08 - 1| = 6.2\%$ compared to DINet and DGNet, respectively. Correspondingly, DIGNet also shows superiority in treatment effect estimation ($\sqrt{\epsilon_{PEHE}}$ and $\epsilon_{ATE}$) compared to DINet (or DGNet), as demonstrated in Table 3.

**Ablation study for PPBR.** *The PPBR approach contributes to enhancing factual outcome predictions, without affecting learning balancing patterns.* From Table 2, we gain two important insights: (1) The difference in learned balancing patterns, measured by $\hat{d}_{\mathcal{H}}$ (or $Wass$), between DINet and INet (or DGNet and GNet), is negligible. This implies that PPBR does not affect learning balancing patterns. (2) Compared with INet, DINet achieves smaller $\sqrt{\epsilon_F}$, with $|1.07/1.08 - 1| = 0.9\%$ error reduction. Similarly, compared with GNet, DGNet achieves smaller $\sqrt{\epsilon_F}$, with $|1.07/1.12 - 1| = 4.5\%$ error reduction. These two observations reveal that PPBR can improve factual outcome predictions, without affecting learning balancing patterns. Benefiting from the advantage of PPBR, the improvement is particularly pronounced in counterfactual estimation. Comparing DINet with INet, the reduction in $\sqrt{\epsilon_{CF}}$ amounts to $|2.95/3.47-1| = 15.0\%$. Similarly, comparing DGNet with GNet, the reduction is $|3.08/3.55 - 1| = 13.2\%$. Correspondingly, DINet (or DGNet) attains smaller treatment effect errors ($\sqrt{\epsilon_{PEHE}}$ and $\epsilon_{ATE}$) compared with INet (or GNet), as shown in Table 3.

Table 3: Training- & test- set $\sqrt{\epsilon_{PEHE}}$ & $\epsilon_{ATE}$ when $\gamma = 3$. Mean ± standard error of 30 runs.

Table 4: Training- & test- set $\sqrt{\epsilon_{PEHE}}$ & $\epsilon_{ATE}$ on IHDP. Mean ± standard error of 100 runs.

| | Training set | | Test set | | | Training set | | Test set | |
|---|---|---|---|---|---|---|---|---|---|
| | $\sqrt{\epsilon_{PEHE}}$ | $\epsilon_{ATE}$ | $\sqrt{\epsilon_{PEHE}}$ | $\epsilon_{ATE}$ | | $\sqrt{\epsilon_{PEHE}}$ | $\epsilon_{ATE}$ | $\sqrt{\epsilon_{PEHE}}$ | $\epsilon_{ATE}$ |
| GNet | 3.30±0.15 | 2.58±0.14 | 3.30±0.16 | 2.59±0.14 | GNet | 0.71±0.15 | 0.12±0.01 | 0.77±0.18 | 0.15±0.02 |
| INet | 3.24±0.11 | 2.46±0.09 | 3.22±0.12 | 2.47±0.10 | INet | 0.66±0.09 | 0.13±0.01 | 0.72±0.11 | 0.15±0.02 |
| DGNet | 2.86±0.06 | 2.15±0.03 | 2.83±0.07 | 2.15±0.04 | DGNet | 0.53±0.07 | **0.11±0.01** | 0.60±0.09 | 0.13±0.01 |
| DINet | 2.70±0.06 | 2.12±0.04 | 2.69±0.08 | 2.13±0.05 | DINet | 0.57±0.12 | 0.13±0.01 | 0.60±0.11 | 0.14±0.01 |
| DIGNet | **2.66±0.07** | **2.04±0.05** | **2.63±0.07** | **2.03±0.04** | DIGNet | **0.42±0.02** | **0.11±0.01** | **0.45±0.04** | **0.12±0.01** |

Table 5: Significance analysis regarding the achieved improvements by comparing GNet and DGNet, INet and DINet, DGNet and DIGNet, DINet and DIGNet. The p-value ≤ 0.05 indicates difference is statistically significant.

| | Training set | | | | Test set | | | |
|---|---|---|---|---|---|---|---|---|
| | $\sqrt{\epsilon_{PEHE}}$ | | $\epsilon_{ATE}$ | | $\sqrt{\epsilon_{PEHE}}$ | | $\epsilon_{ATE}$ | |
| | t-value | p-value | t-value | p-value | t-value | p-value | t-value | p-value |
| GNet vs. DGNet | 2.7435 | **0.0081** | 2.9844 | **0.0042** | 2.7073 | **0.0089** | 2.9269 | **0.0049** |
| INet vs. DINet | 4.0812 | **0.0001** | 3.5222 | **0.0008** | 3.5665 | **0.0007** | 3.0824 | **0.0031** |
| DGNet vs. DIGNet | 2.0240 | **0.0476** | 1.8888 | 0.0639 | 2.0650 | **0.0434** | 2.0935 | **0.0407** |
| DINet vs. DIGNet | 0.4513 | 0.6535 | 1.3525 | 0.1815 | 0.6079 | 0.5456 | 1.5473 | 0.1272 |

**Significance analysis for the improvements.** To assess the significance of the improvements observed in the above ablation studies, we conducted an additional significance analysis by recording the values of $\sqrt{\epsilon_{PEHE}}$ and $\epsilon_{ATE}$ for 30 runs of each of the 5 models (GNet, INet, DGNet, DINet, and DIGNet). Subsequently, we performed a t-test for GNet vs. DGNet, INet vs. DINet, DGNet vs. DIGNet, and DINet vs. DIGNet, to investigate the statistical significance of their differences. The relevant results are reported in Table 5. The results reveal a statistically significant difference between GNet and DGNet, INet and DINet, as well as DGNet and DIGNet. Note that the difference between DINet and DIGNet is not statistically significant, despite DIGNet exhibiting smaller treatment effect estimation errors on average compared to DINet.

### 6.2.3 Comparisons on IHDP benchmark.

In this part, we perform experiments on the IHDP benchmark dataset to compare the performances of different models. The corresponding results are reported in Table 4 and 6.

First, we report the ablation results on 1-100 IHDP datasets in Table 4, aiming to examine the consistent effectiveness of PDIG and PPBR. Specifically, Table 4 shows that DINet and DGNet are superior to INet and GNet but inferior to DIGNet concerning treatment effect estimation, suggesting that both PDIG and PPBR are advantageous for treatment effect estimation. For example, on the test set, DINet reduces $\sqrt{\epsilon_{PEHE}}$ by $|0.60/0.72 - 1| = 16.7\%$ for INet, and DIGNet reduces $\sqrt{\epsilon_{PEHE}}$ by $|0.45/0.60 - 1| = 25\%$ for DINet. This is consistent with the findings before: PDIG and PPBR are beneficial to treatment effect estimation.

Furthermore, we undergo comparisons between DIGNet and other causal models on 1-1000 IHDP datasets and report the results in Table 6. The results highlight the superior performance of the proposed DIGNet across all the models. Specifically, in comparison to the second-best method in test-sample performance, DIGNet achieves a substantial improvement, with error reduced by $|0.45/0.57 - 1| = 21\%$ in terms of $\sqrt{\epsilon_{PEHE}}$ and $|0.12/0.13 - 1| = 7.7\%$ in terms of $\epsilon_{ATE}$. Moreover, it is worth noting that DIGNet consistently achieves the lowest errors across various datasets and metrics, revealing its robust performance. We also conduct an additional experiments on another benchmark dataset Twins. The details and results are deferred to Section A.5

# 7 Conclusion

This paper establishes a theoretical foundation by deriving counterfactual and ITE error bounds based on $\mathcal{H}$-divergence. This theoretical foundation builds a connection between representation balancing and individual propensity confusion. Furthermore, based on individual propensity confusion and group distance minimization, we suggest learning decomposed patterns for representation balancing models using the PDIG and PPBR methods. Further, building upon PDIG and PPBR, we propose a novel model DIGNet, for treatment effect estimation. Comprehensive experiments verify that PDIG and PPBR follow different pathways to

Table 6: Training- & test- set $\sqrt{\epsilon_{PEHE}}$ & $\epsilon_{ATE}$ on IHDP. Mean $\pm$ standard error of 1000 runs.

|  | Training set | | Test set | |
|---|---|---|---|---|
|  | $\sqrt{\epsilon_{PEHE}}$ | $\epsilon_{ATE}$ | $\sqrt{\epsilon_{PEHE}}$ | $\epsilon_{ATE}$ |
| OLS/LR$_1$ (Johansson et al., 2016) | $5.8 \pm .3$ | $.73 \pm .04$ | $5.8 \pm .3$ | $.94 \pm .06$ |
| OLS/LR$_2$ (Johansson et al., 2016) | $2.4 \pm .1$ | $.14 \pm .01$ | $2.5 \pm .1$ | $.31 \pm .02$ |
| k-NN (Crump et al., 2008) | $2.1 \pm .1$ | $.14 \pm .01$ | $4.1 \pm .2$ | $.79 \pm .05$ |
| BART (Chipman et al., 2010) | $2.1 \pm .1$ | $.23 \pm .01$ | $2.3 \pm .1$ | $.34 \pm .02$ |
| CF (Wager & Athey, 2018) | $3.8 \pm .2$ | $.18 \pm .01$ | $3.8 \pm .2$ | $.40 \pm .03$ |
| CEVAE (Louizos et al., 2017) | $2.7 \pm .1$ | $.34 \pm .01$ | $2.6 \pm .1$ | $.46 \pm .02$ |
| SITE (Yao et al., 2018) | $.69 \pm .0$ | $.22 \pm .01$ | $.75 \pm .0$ | $.24 \pm .01$ |
| GANITE (Yoon et al., 2018) | $1.9 \pm .4$ | $.43 \pm .05$ | $2.4 \pm .4$ | $.49 \pm .05$ |
| BLR (Johansson et al., 2016) | $5.8 \pm .3$ | $.72 \pm .04$ | $5.8 \pm .3$ | $.93 \pm .05$ |
| BNN (Johansson et al., 2016) | $2.2 \pm .1$ | $.37 \pm .03$ | $2.1 \pm .1$ | $.42 \pm .03$ |
| TARNet (Shalit et al., 2017) | $.88 \pm .0$ | $.26 \pm .01$ | $.95 \pm .0$ | $.28 \pm .01$ |
| CFR-Wass (GNet) (Shalit et al., 2017) | $.73 \pm .0$ | $.12 \pm .01$ | $.81 \pm .0$ | $.15 \pm .01$ |
| Dragonnet (Shi et al., 2019) | $1.3 \pm .4$ | $.14 \pm .01$ | $1.3 \pm .5$ | $.20 \pm .05$ |
| MBRL (Huang et al., 2022a) | $.52 \pm .0$ | $.12 \pm .01$ | $.57 \pm .0$ | $.13 \pm .01$ |
| DIGNet (Ours) | $\mathbf{.41 \pm .0}$ | $\mathbf{.11 \pm .01}$ | $\mathbf{.46 \pm .0}$ | $\mathbf{.12 \pm .01}$ |

improve counterfactual and ITE estimation. In particular, PDIG enables the model to capture more effective balancing patterns without affecting factual outcome prediction, while PPBR contributes to improving factual outcome predictions without influencing learning balancing patterns. We hope these findings can constitute an important step to inspire more research concerning the generalization of representation balancing models for counterfactual and ITE estimation.

**Limitations and future works.** Our paper verifies the effectiveness of PDIG and PPBR in improving ITE estimation, it is also important to step beyond our empirical insights into future theoretical studies aimed at addressing the trade-off challenge mentioned in the introduction, e.g., exploring the possibility of deriving tighter theoretical error bounds based on learning decomposed patterns, and involving the orthogonal machine learning (Chernozhukov et al., 2018; Oprescu et al., 2019; Nie & Wager, 2021; Huang et al., 2022b) into the representation learning model to improve model's robustness to the misspecification. Furthermore, it remains challenging to analytically determine the best divergence metric for representation balancing methods. A promising avenue for future theoretical investigations would involve developing new distributional divergences or exploring a unified theory that enables models to select appropriate divergence metrics based on the distinct data. Empirical studies can focus on discouraging the redundancy of the concatenation fusion of each decomposed pattern and improving the efficacy of the multi-task learning objectives. While we have followed the same approach as previous studies by evaluating model performance using simulated and semi-synthetic data, it is crucial for future research to explore the creation of appropriate benchmark datasets (Athey & Wager, 2019; Curth et al., 2021) for assessing the performance of ITE estimation methods in real-world scenarios.

# Acknowledgement

We are thankful for the constructive and helpful comments provided by the reviewers and action editor during the reviewing process of TMLR, which has contributed a lot to the improvement of our work.

Qi WU acknowledges the support from The CityU-JD Digits Joint Laboratory in Financial Technology and Engineering and The Hong Kong Research Grants Council [General Research Fund 11219420/9043008 and 11200219/9042900]. The work described in this paper was partially supported by the InnoHK initiative, the Government of the HKSAR, and the Laboratory for AI-Powered Financial Technologies.

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

# A   Appendix

## A.1   Discussion of the Trade-off Problem

We will now discuss two cases to gain a deeper understanding of this trade-off phenomenon. (1) The case without representation balancing: In this case, the outcome functions are fitted by $Y^1 = \hat{\tau}^1(X^{treat})$ and $Y^0 = \hat{\tau}^0(X^{control})$ using treated and controlled samples, respectively. $\hat{\tau}^1(X^{treat})$ and $\hat{\tau}^0(X^{control})$ can be good estimates of factual outcomes based on the well-preserved pre-balancing information (group information). However, the estimated counterfactual outcomes $\hat{\tau}^0(X^{treat})$ and $\hat{\tau}^1(X^{control})$ can be problematic due to the presence of the covariate shift problem $P(X|T=1) \neq P(X|T=0)$, where the distribution of training data $P(X,Y|T=t)$ differs from that of the test data $P(X,Y|T=1-t)$ for $t \in \{0,1\}$ [2]. (2) The case with representation balancing: In this case, the outcome functions are fitted by $Y^1 = \hat{h}^1(\Phi(X^{treat}))$ and $Y^0 = \hat{h}^0(\Phi(X^{control}))$ using treated and controlled samples, respectively. Using $\Phi(X)$ to fit factual outcomes can improve the accuracy of the counterfactual estimates $\hat{h}^0(\Phi(X^{treat}))$ and $\hat{h}^1(\Phi(X^{control}))$, because representation balancing enforces the distributions of treated and controlled samples to be as close as possible in the representation space. As a result, representation balancing effectively tackles the covariate shift issue, resulting in training data and test data following the same distribution [3]. However, executing representation balancing can inevitably lead to a loss of outcome-predictive information in $\Phi(X)$. This occurs naturally as $\Phi$ becomes insensitive to the treatment variable, thereby sacrificing pre-balancing information (group information) that contributes to factual outcome predictions. To illustrate the negative impact of losing pre-balancing information in balanced representations, we present a motivating example below.

**Motivating example.** Suppose there is a vaccine available to prevent a certain disease. We define $X$ as the covariate, $T = 1$ as the treatment (receiving the vaccine), $T = 0$ as the control (not receiving the vaccine), and $Y$ as the outcome (the level of specific antibodies). Assume that the outcome is determined by $Y = T \exp(X) + (1-T) \cdot 0 = T \exp(X)$, which means that if an individual receives the treatment, the level of antibodies will be $y = \exp(x)$; otherwise, it will be $y = 0$. In observational data, the treatment is assigned based on the covariate of each individual. The left graph of Figure 7 illustrates the distributions of $X$ in the treated and control groups. We observe that individuals with positive $x$ values are more likely to receive the vaccine, resulting in a higher level of antibodies. Given some sample $i$, a well trained model first determines whether the sample is more likely to be in the treatment or control group based on its covariate $X_i = x$. If it determines the sample to be in the treatment group, the model then predicts $y = \exp(x)$; otherwise, it predicts $y = 0$. For example, if $x = 1$, the model would classify the sample as more likely to be in the treatment group and predict $y = e$. Therefore, in this case, the pre-balancing covariate remain informative for predicting the outcome. Now, let's consider a representation function $\Phi$ that achieves

---

[2]By unconfoundedness, we have $P(Y|X,T=t) = P(Y|X,T=1-t)$. Due to the covariate shift $P(X|T=t) \neq P(X|T=1-t)$, we have $P(Y|X,T=t)P(X|T=t) \neq P(Y|X,T=1-t)P(X|T=1-t)$, i.e., $P(X,Y|T=t) \neq P(X,Y|T=1-t)$.

[3]By one-to-one and invertible properties of $\Phi$ and unconfoundedness, we have $P(Y|\Phi(X),T=t) = P(Y|\Phi(X),T=1-t)$. Given an ideal representation balancing $P(\Phi(X)|T=t) = P(\Phi(X)|T=1-t)$, we have $P(Y|\Phi(X),T=t)P(\Phi(X)|T=t) = P(Y|\Phi(X),T=1-t)P(\Phi(X)|T=1-t)$, i.e., $P(\Phi(X),Y|T=t) = P(\Phi(X),Y|T=1-t)$.

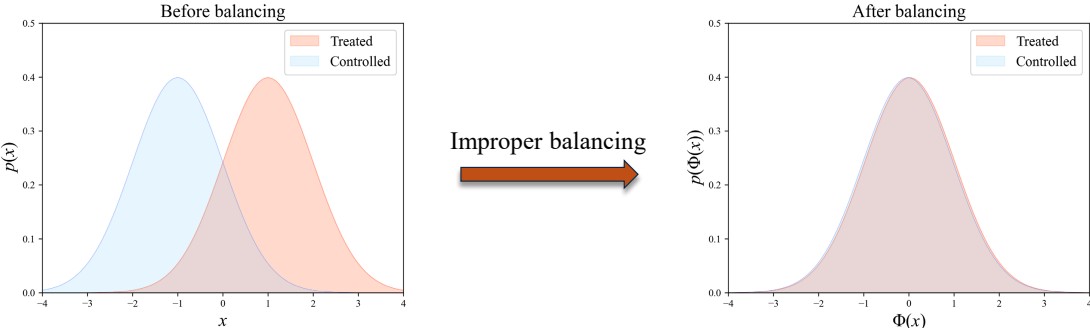

Figure 7: Motivating example (similar to the example demonstrated in Huang et al. (2023)) for illustrating the trade-off between outcome prediction and representation balancing.

improper representation balancing between the treated and control samples. The right graph of Figure 7 shows the distributions of $\Phi(X)$ in the treated and control groups. In this case, it becomes challenging for a model to accurately predict $Y$ using $\Phi(X)$ because the model may become confused about whether a sample is more likely to receive the treatment or the control. Consequently, improperly balanced representations can lead to a loss of outcome-predictive information.

The above discussions and motivating example illustrate the inherent trade-off problem between outcome prediction (Principle I) and representation balancing (Principle II), which arises due to the fact that representation balancing models alleviate covariate shift at the expense of factual outcome prediction.

## A.2 Proof of Lemma 1

Proof of $L$ taking the squared loss, i.e., $L(y_1, y_2) = (y_1 - y_2)^2$:

*Proof.* We denote $\epsilon_{PEHE}(f) = \epsilon_{PEHE}(h, \Phi)$, $\epsilon_F(f) = \epsilon_F(h, \Phi)$, $\epsilon_{CF}(f) = \epsilon_{CF}(h, \Phi)$ for $f(\mathbf{x}, t) = h(\Phi(\mathbf{x}), t)$.

$$
\epsilon_F(f)
$$
$$
= \int_{\mathcal{X} \times \{0,1\} \times \mathcal{Y}} (f(\mathbf{x}, t) - y^t)^2 p(y^t|\mathbf{x}) p(\mathbf{x}, t) dy^t d\mathbf{x} dt
$$
$$
= \int_{\mathcal{X} \times \{0,1\} \times \mathcal{Y}} (f(\mathbf{x}, t) - \tau^t(\mathbf{x}))^2 p(y^t|\mathbf{x}) p(\mathbf{x}, t) dy^t d\mathbf{x} dt
$$
$$
+ \int_{\mathcal{X} \times \{0,1\} \times \mathcal{Y}} (\tau^t(\mathbf{x}) - y^t)^2 p(y^t|\mathbf{x}) p(\mathbf{x}, t) dy^t d\mathbf{x} dt
$$
$$
+ 2 \int_{\mathcal{X} \times \{0,1\} \times \mathcal{Y}} (f(\mathbf{x}, t) - \tau^t(\mathbf{x}))(\tau^t(\mathbf{x}) - y^t) p(y^t|\mathbf{x}) p(\mathbf{x}, t) dy^t d\mathbf{x} dt \tag{28}
$$
$$
= \int_{\mathcal{X} \times \{0,1\}} (f(\mathbf{x}, t) - \tau^t(\mathbf{x}))^2 p(\mathbf{x}, t) d\mathbf{x} dt + \sigma_{y^t}^2(p(\mathbf{x}, t)) \tag{29}
$$

Equation (29) is by the definition of $\sigma_{y^t}^2(p(\mathbf{x}, t))$ in Lemma 1 and equation (28) equaling zero since $\tau^t(\mathbf{x}) = \int_{\mathcal{Y}} y^t p(y^t|\mathbf{x}) dy_t$. A similar result can be obtained for $\epsilon_{CF}$:

$$
\epsilon_{CF}(f) = \int_{\mathcal{X} \times \{0,1\}} (f(\mathbf{x}, t) - \tau^t(\mathbf{x}))^2 p(\mathbf{x}, 1 - t) d\mathbf{x} dt + \sigma_{y^t}^2(p(\mathbf{x}, 1 - t)).
$$

$$\epsilon_{PEHE}(f)$$

$$= \int_{\mathcal{X}} ((f(\mathbf{x}, 1) - f(\mathbf{x}, 0)) - (\tau^1(\mathbf{x}) - \tau^0(\mathbf{x})))^2 p(\mathbf{x}) d\mathbf{x}$$

$$\leq 2 \int_{\mathcal{X}} ((f(\mathbf{x}, 1) - \tau^1(\mathbf{x}))^2 + (f(\mathbf{x}, 0) - \tau^0(\mathbf{x}))^2) p(\mathbf{x}) d\mathbf{x} \tag{30}$$

$$= 2 \int_{\mathcal{X}} (f(\mathbf{x}, 1) - \tau^1(\mathbf{x}))^2 p(\mathbf{x}, T = 1) d\mathbf{x} + 2 \int_{\mathcal{X}} (f(\mathbf{x}, 0) - \tau^0(\mathbf{x}))^2 p(\mathbf{x}, T = 0) d\mathbf{x}$$

$$+ 2 \int_{\mathcal{X}} (f(\mathbf{x}, 1) - \tau^1(\mathbf{x}))^2 p(\mathbf{x}, T = 0) d\mathbf{x} + 2 \int_{\mathcal{X}} (f(\mathbf{x}, 0) - \tau^0(\mathbf{x}))^2 p(\mathbf{x}, T = 1) d\mathbf{x} \tag{31}$$

$$= 2 \int_{\mathcal{X} \times \{0,1\}} (f(\mathbf{x}, t) - \tau^t(\mathbf{x}))^2 p(\mathbf{x}, t) d\mathbf{x} dt + 2 \int_{\mathcal{X} \times \{0,1\}} (f(\mathbf{x}, t) - \tau^t(\mathbf{x}))^2 p(\mathbf{x}, 1 - t) d\mathbf{x} dt$$

$$= 2(\epsilon_F(f) - \sigma_{y^t}^2(p(\mathbf{x}, t))) + 2(\epsilon_{CF}(f) - \sigma_{y^t}^2(p(\mathbf{x}, 1 - t))). \tag{32}$$

Inequality (30) is by $(x + y)^2 \leq 2(x^2 + y^2)$; equation (31) is by $p(\mathbf{x}) = p(\mathbf{x}, T = 0) + p(\mathbf{x}, T = 1)$. By (equation 32) and the definition of $\sigma_y^2$ in Lemma 1, we have

$$\epsilon_{PEHE}(h, \Phi) \leq 2(\epsilon_{CF}(h, \Phi) + \epsilon_F(h, \Phi) - 2\sigma_y^2).$$

$\square$

Proof of $L$ taking the absolute loss, i.e., $L(y_1, y_2) = |y_1 - y_2|$:

*Proof.* We denote $\epsilon_{PEHE}(f) = \epsilon_{PEHE}(h, \Phi)$, $\epsilon_F(f) = \epsilon_F(h, \Phi)$, $\epsilon_{CF}(f) = \epsilon_{CF}(h, \Phi)$ for $f(\mathbf{x}, t) = h(\Phi(\mathbf{x}), t)$.

$$\epsilon_F(f)$$

$$= \int_{\mathcal{X} \times \{0,1\} \times \mathcal{Y}} |f(\mathbf{x}, t) - y^t| p(y^t | \mathbf{x}) p(\mathbf{x}, t) dy^t d\mathbf{x} dt$$

$$\geq \int_{\mathcal{X} \times \{0,1\} \times \mathcal{Y}} |f(\mathbf{x}, t) - \tau^t(\mathbf{x})| p(y^t | \mathbf{x}) p(\mathbf{x}, t) dy^t d\mathbf{x} dt$$

$$- \int_{\mathcal{X} \times \{0,1\} \times \mathcal{Y}} |\tau^t(\mathbf{x}) - y^t| p(y^t | \mathbf{x}) p(\mathbf{x}, t) dy^t d\mathbf{x} dt \tag{33}$$

$$= \int_{\mathcal{X} \times \{0,1\}} |f(\mathbf{x}, t) - \tau^t(\mathbf{x})| p(\mathbf{x}, t) d\mathbf{x} dt - A_{y^t}(p(\mathbf{x}, t)). \tag{34}$$

Inequality (33) is by $|x + y| \geq |x| - |y|$, equation (34) is by the definition of $A_{y^t}(p(\mathbf{x}, t))$ in Lemma 1. A similar result can be obtained for $\epsilon_{CF}$:

$$\epsilon_{CF}(f) \geq \int_{\mathcal{X} \times \{0,1\}} |f(\mathbf{x}, t) - \tau^t(\mathbf{x})| p(\mathbf{x}, 1 - t) d\mathbf{x} dt - A_{y^t}(p(\mathbf{x}, 1 - t)).$$

$$\epsilon_{PEHE}(f)$$

$$= \int_{\mathcal{X}} |(f(\mathbf{x}, 1) - f(\mathbf{x}, 0)) - (\tau^1(\mathbf{x}) - \tau^0(\mathbf{x}))| p(\mathbf{x}) d\mathbf{x}$$

$$\leq \int_{\mathcal{X}} (|f(\mathbf{x}, 1) - \tau^1(\mathbf{x})| + |f(\mathbf{x}, 0) - \tau^0(\mathbf{x})|) p(\mathbf{x}) d\mathbf{x} \tag{35}$$

$$= \int_{\mathcal{X}} |f(\mathbf{x}, 1) - \tau^1(\mathbf{x})| p(\mathbf{x}, T = 1) d\mathbf{x} + \int_{\mathcal{X}} |f(\mathbf{x}, 1) - \tau^1(\mathbf{x})| p(\mathbf{x}, T = 0) d\mathbf{x} \tag{36}$$

$$+ \int_{\mathcal{X}} |f(\mathbf{x}, 0) - \tau^0(\mathbf{x})| p(\mathbf{x}, T = 0) d\mathbf{x} + \int_{\mathcal{X}} |f(\mathbf{x}, 0) - \tau^0(\mathbf{x})| p(\mathbf{x}, T = 1) d\mathbf{x} \tag{37}$$

$$= \int_{\mathcal{X} \times \{0,1\}} |f(\mathbf{x}, t) - \tau^t(\mathbf{x})| p(\mathbf{x}, t) d\mathbf{x} dt + \int_{\mathcal{X} \times \{0,1\}} |f(\mathbf{x}, t) - \tau^t(\mathbf{x})| p(\mathbf{x}, 1 - t) d\mathbf{x} dt$$

$$\leq \epsilon_F(f) + A_{y^t}(p(\mathbf{x}, t)) + \epsilon_{CF}(f) + A_{y^t}(p(\mathbf{x}, 1 - t)). \tag{38}$$

Inequality (35) is by $|x + y| \leq |x| + |y|$. Equation (36) and equation (37) are by $p(\mathbf{x}) = p(\mathbf{x}, T = 0) + p(\mathbf{x}, T = 1)$. By equation (38) and the definition of $A_y$ in Lemma 1, we have

$$\epsilon_{PEHE}(h, \Phi) \leq \epsilon_F(f) + A_{y^t}(p(\mathbf{x}, t)) + \epsilon_{CF}(f) + A_{y^t}(p(\mathbf{x}, 1 - t))$$
$$\leq \epsilon_{CF}(h, \Phi) + \epsilon_F(h, \Phi) + 2A_y.$$

$$\square$$

### A.3 Proof of Theorem 1

Proof of equation (4):

*Proof.*

$$\epsilon_{CF}(h, \Phi) - [(1 - u) \cdot \epsilon_F^{T=1}(h, \Phi) + u \cdot \epsilon_F^{T=0}(h, \Phi)]$$

$$= [(1 - u) \cdot \epsilon_{CF}^{T=1}(h, \Phi) + u \cdot \epsilon_{CF}^{T=0}(h, \Phi)] - [(1 - u) \cdot \epsilon_F^{T=1}(h, \Phi) + u \cdot \epsilon_F^{T=0}(h, \Phi)]$$

$$= (1 - u) \cdot [\epsilon_{CF}^{T=1}(h, \Phi) - \epsilon_F^{T=1}(h, \Phi)] + u \cdot [\epsilon_{CF}^{T=0}(h, \Phi) - \epsilon_F^{T=0}(h, \Phi)]$$

$$= (1 - u) \int_{\mathcal{X}} \ell_{h,\Phi}(\mathbf{x}, 1)(p^{T=0}(\mathbf{x}) - p^{T=1}(\mathbf{x})) d\mathbf{x} + u \int_{\mathcal{X}} \ell_{h,\Phi}(\mathbf{x}, 0)(p^{T=1}(\mathbf{x}) - p^{T=0}(\mathbf{x})) d\mathbf{x}$$

$$= (1 - u) \int_{\mathcal{R}} \ell_{h,\Phi}(\Psi(\mathbf{r}), 1)(p_\Phi^{T=0}(\mathbf{r}) - p_\Phi^{T=1}(\mathbf{r})) d\mathbf{r} + u \int_{\mathcal{R}} \ell_{h,\Phi}(\Psi(\mathbf{r}), 0)(p_\Phi^{T=1}(\mathbf{r}) - p_\Phi^{T=0}(\mathbf{r})) d\mathbf{r} \tag{39}$$

$$= B_\Phi \cdot (1 - u) \int_{\mathcal{R}} \frac{1}{B_\Phi} \ell_{h,\Phi}(\Psi(\mathbf{r}), 1)(p_\Phi^{T=0}(\mathbf{r}) - p_\Phi^{T=1}(\mathbf{r})) d\mathbf{r}$$

$$+ B_\Phi \cdot u \int_{\mathcal{R}} \frac{1}{B_\Phi} \ell_{h,\Phi}(\Psi(\mathbf{r}), 0)(p_\Phi^{T=1}(\mathbf{r}) - p_\Phi^{T=0}(\mathbf{r})) d\mathbf{r}$$

$$\leq B_\Phi \cdot (1 - u) \sup_{g \in \mathcal{G}} |\int_{\mathcal{R}} g(\mathbf{r})(p_\Phi^{T=0}(\mathbf{r}) - p_\Phi^{T=1}(\mathbf{r})) d\mathbf{r}|$$

$$+ B_\Phi \cdot u \cdot \sup_{g \in \mathcal{G}} |\int_{\mathcal{R}} g(\mathbf{r})(p_\Phi^{T=1}(\mathbf{r}) - p_\Phi^{T=0}(\mathbf{r})) d\mathbf{r}| \tag{40}$$

$$= B_\Phi \cdot Wass(p_\Phi^{T=1}, p_\Phi^{T=0}) \tag{41}$$

Equation (39) is by the change of formula, $p_\Phi^{T=0}(\mathbf{r}) = p^{T=0}(\Psi(\mathbf{r})) J_\Psi(\mathbf{r})$, $p_\Phi^{T=1}(\mathbf{r}) = p^{T=1}(\Psi(\mathbf{r})) J_\Psi(\mathbf{r})$, where $J_\Psi(\mathbf{r})$ is the absolute of the determinant of the Jacobian of $\Psi(\mathbf{r})$. Equation (41) is by Definition 2. $\square$

Proof of equation (5):

*Proof.*

$$\epsilon_{PEHE}(h, \Phi)$$
$$\leq 2(\epsilon_{CF}(h, \Phi) + \epsilon_F(h, \Phi) - 2\sigma_y^2). \tag{42}$$
$$\leq 2(\epsilon_F^{T=1}(h, \Phi) + \epsilon_F^{T=0}(h, \Phi) + B_\Phi \cdot Wass(p_\Phi^{T=1}, p_\Phi^{T=0}) - 2\sigma_y^2). \tag{43}$$

Inequality (42) is by equation (2) in Lemma 1. Inequality (43) is by equation 4 in Theorem 1. □

Proof of equation (6):

*Proof.*

$$\epsilon_{PEHE}(h, \Phi)$$
$$\leq \epsilon_{CF}(h, \Phi) + \epsilon_F(h, \Phi) + 2A_y \tag{44}$$
$$\leq \epsilon_F^{T=1}(h, \Phi) + \epsilon_F^{T=0}(h, \Phi) + B_\Phi \cdot Wass(p_\Phi^{T=1}, p_\Phi^{T=0}) + 2A_y \tag{45}$$

Inequality (44) is by equation (3) in Lemma 1. Inequality (45) is by equation 4 in Theorem 1. □

## A.4 Proof of Theorem 2

We first introduce Lemma 2 that is useful for proving Theorem 2.

**Lemma 2.** *Let $\mathcal{G}$ that is defined in Definition 2 be the family of binary functions. Then we obtain* $\sup_{\eta \in \mathcal{H}} \left| \int_{\mathcal{S}} \eta(s)(p_1(s) - p_2(s))ds \right| = \frac{1}{2}d_{\mathcal{H}}(p_1, p_2).$

*Proof.* Let $\mathbb{I}(\cdot)$ denotes an indicator function.

$$d_{\mathcal{H}}(p_1, p_2)$$
$$= 2 \sup_{\eta \in \mathcal{H}} \left| \int_{\eta(s)=1} (p_1(s) - p_2(s))ds \right|$$
$$= 2 \sup_{\eta \in \mathcal{H}} \left| \int_{\mathcal{S}} \mathbb{I}(\eta(s) = 1)(p_1(s) - p_2(s))ds \right|$$
$$= 2 \sup_{\eta \in \mathcal{H}} \left| \int_{\mathcal{S}} \eta(s)(p_1(s) - p_2(s))ds \right| \tag{46}$$

The last equation is because an indicator function is also a binary function. □

Proof of equation (8):

*Proof.*

$$\epsilon_{CF}(h, \Phi) - [(1 - u) \cdot \epsilon_F^{T=1}(h, \Phi) + u \cdot \epsilon_F^{T=0}(h, \Phi)]$$

$$=(1 - u) \int_{\mathcal{R}} \ell_{h,\Phi}(\Psi(\mathbf{r}), 1)(p_\Phi^{T=0}(\mathbf{r}) - p_\Phi^{T=1}(\mathbf{r}))d\mathbf{r} + u \int_{\mathcal{R}} \ell_{h,\Phi}(\Psi(\mathbf{r}), 0)(p_\Phi^{T=1}(\mathbf{r}) - p_\Phi^{T=0}(\mathbf{r}))d\mathbf{r} \quad (47)$$

$$\leq(1 - u) \int_{p_\Phi^{T=0} > p_\Phi^{T=1}} \ell_{h,\Phi}(\Psi(\mathbf{r}), 1)(p_\Phi^{T=0}(\mathbf{r}) - p_\Phi^{T=1}(\mathbf{r}))d\mathbf{r}$$

$$+ u \int_{p_\Phi^{T=1} > p_\Phi^{T=0}} \ell_{h,\Phi}(\Psi(\mathbf{r}), 0)(p_\Phi^{T=1}(\mathbf{r}) - p_\Phi^{T=0}(\mathbf{r}))d\mathbf{r} \quad (48)$$

$$\leq(1 - u)K \int_{p_\Phi^{T=0} > p_\Phi^{T=1}} (p_\Phi^{T=0}(\mathbf{r}) - p_\Phi^{T=1}(\mathbf{r}))d\mathbf{r} + u \cdot K \int_{p_\Phi^{T=1} > p_\Phi^{T=0}} (p_\Phi^{T=1}(\mathbf{r}) - p_\Phi^{T=0}(\mathbf{r}))d\mathbf{r} \quad (49)$$

$$=(1 - u)K \int_{\mathcal{R}} \mathbb{I}(p_\Phi^{t=0} > p_\Phi^{T=1})(p_\Phi^{T=0}(\mathbf{r}) - p_\Phi^{T=1}(\mathbf{r}))d\mathbf{r}$$

$$+ u \cdot K \int_{\mathcal{R}} \mathbb{I}(p_\Phi^{T=1} > p_\Phi^{T=0})(p_\Phi^{T=1}(\mathbf{r}) - p_\Phi^{T=0}(\mathbf{r}))d\mathbf{r}$$

$$\leq(1 - u)K \sup_{\eta \in \mathcal{H}} | \int_{\mathcal{R}} \eta(\mathbf{r})(p_\Phi^{T=1}(\mathbf{r}) - p_\Phi^{T=0}(\mathbf{r}))d\mathbf{r}|$$

$$+ u \cdot K \cdot \sup_{\eta \in \mathcal{H}} | \int_{\mathcal{R}} \eta(\mathbf{r})(p_\Phi^{T=1}(\mathbf{r}) - p_\Phi^{T=0}(\mathbf{r}))d\mathbf{r}| \quad (50)$$

$$\leq K \cdot \sup_{\eta \in \mathcal{H}} | \int_{\mathcal{R}} \eta(\mathbf{r})((p_\Phi^{T=1}(\mathbf{r}) - p_\Phi^{T=0}(\mathbf{r})))d\mathbf{r}|$$

$$=\frac{K}{2} d_{\mathcal{H}}(p_\Phi^{T=1}, p_\Phi^{T=0}) \quad (51)$$

Equation (47) is derived in the same way as equation (39). Equation (48) is by $\ell_{h,\Phi} \geq 0$ for all $\mathbf{r}$ and $t$. Inequality (49) is by the definition of $K$ in Theorem 2. Inequality (50) is because an indicator function is also a binary function. Equation (51) is by Lemma 2. $\square$

Proof of equation (9):

*Proof.*

$$\epsilon_{PEHE}(h, \Phi)$$

$$\leq 2(\epsilon_{CF}(h, \Phi) + \epsilon_F(h, \Phi) - 2\sigma_y^2) \quad (52)$$

$$\leq 2(\epsilon_F^{T=1}(h, \Phi) + \epsilon_F^{T=0}(h, \Phi) + \frac{K}{2} d_{\mathcal{H}}(p_\Phi^{T=1}, p_\Phi^{T=0}) - 2\sigma_y^2) \quad (53)$$

Inequality (52) is by equation 2 in Lemma 1. Inequality (53) is by equation 8 in Theorem 2. $\square$

Proof of equation (10):

*Proof.*

$$\epsilon_{PEHE}(h, \Phi)$$

$$\leq \epsilon_{CF}(h, \Phi) + \epsilon_F(h, \Phi) + 2A_y \quad (54)$$

$$\leq \epsilon_F^{T=1}(h, \Phi) + \epsilon_F^{T=0}(h, \Phi) + \frac{K}{2} d_{\mathcal{H}}(p_\Phi^{T=1}, p_\Phi^{T=0}) + 2A_y \quad (55)$$

Inequality (54) is by equation 3 in Lemma 1. Inequality (55) is by equation 8 in Theorem 2. $\square$

## A.5 Additional Experimental details

**Additional results on Twins Benchmark.** To investigate the applicability of our model DIGNet to benchmark datasets beyond the commonly used IHDP benchmark, we conducted additional comparisons with several baseline models, including linear, tree, matching, and representation learning methods, on the Twins benchmark, as presented in Table 7.

The Twins dataset comprises records of twin births in the USA between 1989 and 1991. After preprocessing, each unit contains 30 covariates relevant to parents, pregnancy, and birth. The treatment $D = 1$ indicates the heavier twin, while $D = 0$ indicates the lighter twin. The binary outcome variable $Y$ represents 1-year mortality. For more comprehensive details on this dataset and the limitation of IHDP, refer to Curth et al. (2021).

Notably, for $\epsilon_{ATE}$, the simple linear or matching estimator performs best across different methods. On the other hand, when assessing ITE performance using the AUC of potential outcomes, representation learning models all demonstrate strong performance, with AUC values exceeding 0.800 on both training and test sets. The observation might stem from the fact that representation balancing models are based on ITE error bounds, rather than ATE error bounds, thereby optimizing for AUC instead of $\epsilon_{ATE}$. Moreover, among all the models, our DIGNet achieves the second-best AUC results. The best results are achieved by MBRL, which involves the orthogonality information (similar to doubly robust estimators) in representation balancing. This, in turn, inspires us to explore ATE error bounds, or consider involving doubly robust methods in future research.

Table 7: Training- & test- set AUC & $\epsilon_{ATE}$ on Twins. Mean $\pm$ standard error of 100 runs.

| | Training set | | Test set | |
|---|---|---|---|---|
| | AUC | $\epsilon_{ATE}$ | AUC | $\epsilon_{ATE}$ |
| OLS/LR$_1$ Johansson et al. (2016) | .660 ± .005 | .004 ± .003 | .500 ± .028 | .007 ± .006 |
| OLS/LR$_2$ Johansson et al. (2016) | .660 ± .004 | .004 ± .003 | .500 ± .016 | .007 ± .006 |
| k-NN Crump et al. (2008) | .609 ± .010 | .003 ± .002 | .492 ± .012 | .005 ± .004 |
| BART Chipman et al. (2010) | .506 ± .014 | .121 ± .024 | .500 ± .011 | .127 ± .024 |
| CEVAE Louizos et al. (2017) | .845 ± .003 | .022 ± .002 | .841 ± .004 | .032 ± .003 |
| SITE Yao et al. (2018) | .862 ± .002 | .016 ± .001 | .853 ± .006 | .020 ± .002 |
| BLR Johansson et al. (2016) | .611 ± .009 | .006 ± .004 | .510 ± .018 | .033 ± .009 |
| BNN Johansson et al. (2016) | .690 ± .008 | .006 ± .003 | .676 ± .008 | .020 ± .007 |
| TARNet Shalit et al. (2017) | .849 ± .002 | .011 ± .002 | .840 ± .006 | .015 ± .002 |
| CFR-Wass (GNet) Shalit et al. (2017) | .850 ± .002 | .011 ± .002 | .842 ± .005 | .028 ± .003 |
| MBRL (Huang et al., 2022a) | .879 ± .000 | .003 ± .000 | .874 ± .001 | .007 ± .00q |
| DIGNet (Ours) | .874 ± .001 | .004 ± .001 | .871 ± .001 | .008 ± .001 |

**Implementation details.** In simulation studies, we ensure a fair comparison by fixing all the hyperparameters in all datasets across different models. The relevant details are stated in Table 8. In IHDP studies,

Table 8: Hyperparameters of different models in simulation studies.

| | $\Phi_E$ | $\Phi_G$ | $\Phi_I$ | $\pi$ | $h^1$ | $h^0$ | $\alpha_1$ | $\alpha_2$ | batchsize | iteration | learning rate | learning rate for $\pi$ |
|---|---|---|---|---|---|---|---|---|---|---|---|---|
| Gnet | (100, 100, 100, 100) | − | − | − | (100, 100) | (100, 100) | 0.1 | − | 100 | 300 | $1e^{-3}$ | − |
| Inet | (100, 100, 100, 100) | − | − | (100, 100, 100) | (100, 100) | (100, 100) | − | 0.1 | 100 | 300 | $1e^{-3}$ | $1e^{-4}$ |
| DGNet | (100, 100, 100, 100) | (100, 100) | − | − | (100, 100) | (100, 100) | 0.1 | − | 100 | 300 | $1e^{-3}$ | − |
| DINet | (100, 100, 100, 100) | − | (100, 100) | (100, 100, 100) | (100, 100) | (100, 100) | − | 0.1 | 100 | 300 | $1e^{-3}$ | $1e^{-4}$ |
| DIGNet | (100, 100, 100, 100) | (100, 100) | (100, 100) | (100, 100, 100) | (100, 100) | (100, 100) | 0.1 | 0.1 | 100 | 300 | $1e^{-3}$ | $1e^{-4}$ |

to compare with the baseline model CFR-Wass (GNet), we remain the hyperparameters of INet, DGNet, DINet and the early stopping rule the same as those used in CFR-Wass Shalit et al. (2017). Since DIGNet

is more complex than other four models, we adjust the hyperparameters of $\Phi_E$, $\Phi_G$, $\Phi_I$, $\alpha_1$, and $\alpha_2$ for DIGNet as Shalit et al. (2017) do. The relevant details are stated in Table 9.

Table 9: Hyperparameters of different models in IHDP experiments.

| | $\Phi_E$ | $\Phi_G$ | $\Phi_I$ | $\pi$ | $h^1$ | $h^0$ | $\alpha_1$ | $\alpha_2$ | batchsize | iteration | learning rate | learning rate for $\pi$ |
|---|---|---|---|---|---|---|---|---|---|---|---|---|
| Gnet | $(100,100,100,100)$ | – | – | – | $(100,100,100)$ | $(100,100,100)$ | 1 | – | 100 | 600 | $1e^{-3}$ | – |
| Inet | $(100,100,100,100)$ | – | – | $(200,200,200)$ | $(100,100,100)$ | $(100,100,100)$ | – | 1 | 100 | 600 | $1e^{-3}$ | $1e^{-3}$ |
| DGNet | $(100,100,100,100)$ | $(100,100)$ | – | – | $(100,100,100)$ | $(100,100,100)$ | 1 | – | 100 | 600 | $1e^{-3}$ | – |
| DINet | $(100,100,100,100)$ | – | $(100,100)$ | $(200,200,200)$ | $(100,100,100)$ | $(100,100,100)$ | – | 1 | 100 | 600 | $1e^{-3}$ | $1e^{-3}$ |
| DIGNet | $(100,100,100,100,100,100)$ | $(100,100,100)$ | $(100,100,100)$ | $(200,200,200)$ | $(100,100,100)$ | $(100,100,100)$ | 1 | 1 | 100 | 600 | $1e^{-3}$ | $1e^{-3}$ |

**Analysis of training time and training stability.** We record the time it took for different models to run through 100 IHDP datasets in Table 10, and each model is trained within 600 epochs. Following Shalit et al. (2017), all models adopt the early stopping rule. We also record the average early stopping epoch on 100 runs and the actual time on 100 runs, where (actual time) = (total time) × (average early stopping epoch)/600. Not surprisingly, GNet took the least amount of time with 3096 seconds since the objective of GNet is the simplest. However, it is very interesting that the proposed methods, DGNet and DINet, are the first two to early stop. As a result, though DGNet and DINet have multi-objectives, they spent less actual training time but achieved better ITE estimation compared to GNet and INet. Since GNet and INet are actually DGNet and DINet with PPBR ablated, we find that PPBR component can help a model achieve better ITE estimates with less time. In addition, we find that DIGNet spent the longest time to optimize since it has the most complex objective. To further study the stability of the model training, we also plot the metrics $\sqrt{\epsilon_F}$, Wass, $\hat{d}_{\mathcal{H}}$, and $\sqrt{\epsilon_{PEHE}}$ for the first 100 epochs of each model on the first IHDP dataset in Figure 8. We find that the training process of DIGNet is stable, even steadier than GNet and INet. From this perspective, we haven't seen a difficulty of optimizing DIGNet.

Table 10: Training time records on 100 IHDP datasets.

| Model | Time for 600 epochs | Avg early stopping | Actual time | $\sqrt{\epsilon_{PEHE}}$ on test set |
|---|---|---|---|---|
| GNet | 3096s | 240.61 | 1241s | $0.77 \pm 0.18$ |
| INet | 4042s | 254.19 | 1712s | $0.72 \pm 0.11$ |
| DGNet | 3775s | 169.17 | 1064s | $0.60 \pm 0.09$ |
| DINet | 3212s | 157.98 | 846s | $0.60 \pm 0.11$ |
| DIGNet | 4984s | 226.76 | 1884s | $0.45 \pm 0.04$ |

We also provide the ITE and ATE estimation results on 100 IHDP datasets when the combination of $(\alpha_1, \alpha_2)$ in DIGNet objective varies in $\{0.1, 0.5, 1\}$. The relevant results are reported in Table 11, indicating our DIGNet model is robust to the hyperparameters varying.

Table 11: The results on 100 IHDP datasets with different combinations of $(\alpha_1, \alpha_2)$ in DIGNet objective.

| | Training set | | Test set | |
|---|---|---|---|---|
| $(\alpha_1, \alpha_2)$ | $\sqrt{\epsilon_{PEHE}}$ | $\epsilon_{ATE}$ | $\sqrt{\epsilon_{PEHE}}$ | $\epsilon_{ATE}$ |
| $(0.1, 0.1)$ | $0.407 \pm 0.018$ | $0.125 \pm 0.015$ | $0.434 \pm 0.022$ | $0.138 \pm 0.016$ |
| $(0.1, 0.5)$ | $0.414 \pm 0.026$ | $0.120 \pm 0.015$ | $0.434 \pm 0.028$ | $0.123 \pm 0.015$ |
| $(0.1, 1)$ | $0.416 \pm 0.019$ | $0.116 \pm 0.014$ | $0.452 \pm 0.026$ | $0.121 \pm 0.015$ |
| $(0.5, 0.1)$ | $0.417 \pm 0.023$ | $0.130 \pm 0.016$ | $0.440 \pm 0.026$ | $0.137 \pm 0.017$ |
| $(0.5, 0.5)$ | $0.407 \pm 0.021$ | $0.125 \pm 0.015$ | $0.416 \pm 0.022$ | $0.124 \pm 0.015$ |
| $(0.5, 1)$ | $0.413 \pm 0.020$ | $0.126 \pm 0.014$ | $0.455 \pm 0.028$ | $0.133 \pm 0.016$ |
| $(1, 0.1)$ | $0.411 \pm 0.021$ | $0.119 \pm 0.015$ | $0.439 \pm 0.027$ | $0.118 \pm 0.015$ |
| $(1, 0.5)$ | $0.403 \pm 0.020$ | $0.118 \pm 0.015$ | $0.430 \pm 0.026$ | $0.128 \pm 0.016$ |
| $(1, 1)$ | $0.402 \pm 0.019$ | $0.112 \pm 0.014$ | $0.437 \pm 0.027$ | $0.121 \pm 0.015$ |

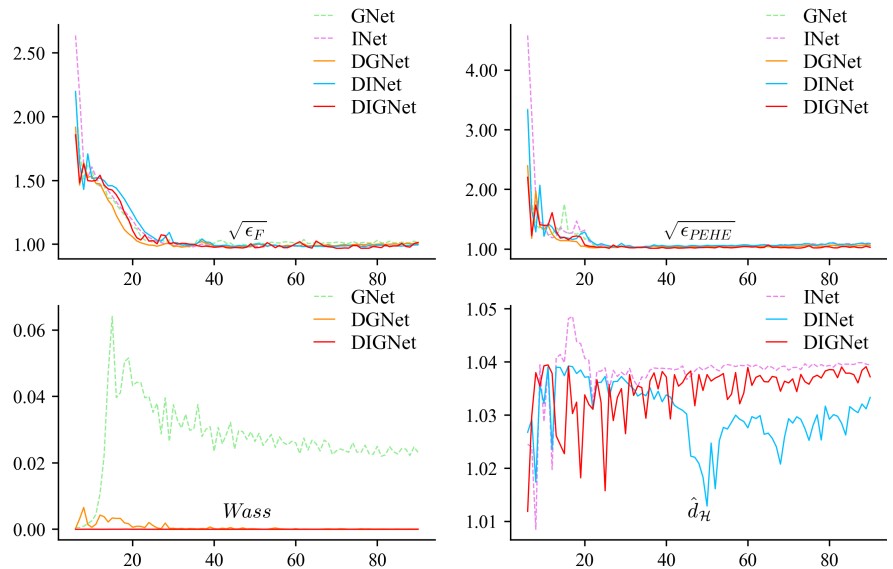

Figure 8: Training loss plots for the first 100 epochs on the first IHDP dataset.

### A.6   Objectives of Different Models

**Objective of GNet.**

$$\min_{\Phi_E, h^t} \quad \mathcal{L}_y(\mathbf{x}, \mathbf{t}, \mathbf{y}; \Phi_E, h^t) + \alpha_1 \mathcal{L}_G(\mathbf{x}, \mathbf{t}; \Phi_E).$$

**Objective of INet.**

$$\max_{\pi} \quad \alpha_2 \mathcal{L}_I(\mathbf{x}, \mathbf{t}; \Phi_E, \pi),$$
$$\min_{\Phi_E, h^t} \quad \mathcal{L}_y(\mathbf{x}, \mathbf{t}, \mathbf{y}; \Phi_E, h^t) + \alpha_2 \mathcal{L}_I(\mathbf{x}, \mathbf{t}; \Phi_E, \pi).$$

**Objective of DINet.**   Note that similar to DIGNet, the pre-balancing patterns are preserved by only updating $\Phi_I$ but fixing $\Phi_E$ in the second step.

$$\max_{\pi} \quad \alpha_2 \mathcal{L}_I(\mathbf{x}, \mathbf{t}; \Phi_I \circ \Phi_E, \pi),$$
$$\min_{\Phi_I} \quad \alpha_2 \mathcal{L}_I(\mathbf{x}, \mathbf{t}; \Phi_I \circ \Phi_E, \pi),$$
$$\min_{\Phi_E, \Phi_I, h^t} \quad \mathcal{L}_y(\mathbf{x}, \mathbf{t}, \mathbf{y}; \Phi_E \oplus (\Phi_I \circ \Phi_E), h^t).$$

**Objective of DGNet.**   Note that similar to DIGNet, the pre-balancing patterns are preserved by only updating $\Phi_G$ but fixing $\Phi_E$ in the first step.

$$\min_{\Phi_G} \quad \alpha_1 \mathcal{L}_G(\mathbf{x}, \mathbf{t}; \Phi_G \circ \Phi_E),$$
$$\min_{\Phi_E, \Phi_G, h^t} \quad \mathcal{L}_y(\mathbf{x}, \mathbf{t}, \mathbf{y}; \Phi_E \oplus (\Phi_G \circ \Phi_E), h^t).$$

**Objective of DIGNet.**

$$\min_{\Phi_G} \quad \alpha_1 \mathcal{L}_G(\mathbf{x}, \mathbf{t}; \Phi_G \circ \Phi_E),$$

$$\max_{\pi} \quad \alpha_2 \mathcal{L}_I(\mathbf{x}, \mathbf{t}; \Phi_I \circ \Phi_E, \pi),$$

$$\min_{\Phi_I} \quad \alpha_2 \mathcal{L}_I(\mathbf{x}, \mathbf{t}; \Phi_I \circ \Phi_E, \pi),$$

$$\min_{\Phi_E, \Phi_I, \Phi_G, h^t} \quad \mathcal{L}_y(\mathbf{x}, \mathbf{t}, \mathbf{y}; \Phi_E \oplus (\Phi_I \circ \Phi_E) \oplus (\Phi_G \circ \Phi_E), h^t).$$

