# OpenReview forum: "DIGNet: Learning Decomposed Patterns in Representation Balancing for Treatment Effect Estimation"
_TMLR — Accepted by TMLR_

### Review · Reviewer_2zS4 · 2024-02-25

**Summary Of Contributions:**

This paper studies learning representation for treatment effect estimation under covariate shift. This paper proposes a representation balancing model, called DIGNet, for treatment effort estimation under covariate shift.

**Audience:**

Yes

**Broader Impact Concerns:**

No concerns

**Claims And Evidence:**

Yes

**Requested Changes:**

- This paper is about approximation rather than estimation. Given sufficient data samples, it is possible to estimate causal effects under a covariate shift. I assume this paper works on a numerical approximation from samples.
- The motivation example is confusing as the age is identical to the treatment. It would be better to use a probabilistic relationship between - In Sec 2 preliminaries, there are the N iid random variables --> N randomly selected samples. The factual outcome and counterfactual come should be distinct.
- In the discussion in Theorem 2, "This new theoretical result provides a theoretical foundation for representation balancing models based on individual propensity confusion." Please explain either in this paragraph or in Sec 4.1.2.

**Strengths And Weaknesses:**

**Strengths**
- This paper develops upper bounds for the counterfactual errors and ITE errors based on H-divergence.
- The proposed framework DIGNet learns representations by adversarial learning using a set of objectives to balance factual prediction and causal effect estimation.

**Weaknesses**
- The motivation is not clear. It seems this paper targets treatment effect estimation using representation balancing. The underlying mechanism reveals the factual errors and the distribution discrepancy bound the counterfactual error. But the contradiction is unclear. I wonder whether one can directly optimize Eq (4).
- The framework is based on potential outcomes and adopts the most common assumptions in potential outcomes. However, the framework implicitly assumes X->T->Y and X -> Y.
- The proposed framework targets to mitigate the trade-off problem. However, the objective functions in Eq (20-23) are still difficult to balance. The adversarial games and the coefficients alpha are both hard to handle in optimization.
- The method is only evaluated in synthetic data and semi-synthetic data. It is not evaluated in real-world datasets. The real-world datasets may not have the ground truth but there might be some challenges, such as causal graphs, and hyper-parameter tuning. I wonder whether the proposed method is sensitive or robust for misspecification.

---

### Review · Reviewer_eHay · 2024-02-26

**Summary Of Contributions:**

This paper presents an approach for estimating individual treatment effects - given an input X, the difference between the treatment and control potential (i.e. counterfactual) outcomes. The approach relies on two techniques for removing information from embeddings which have been explored in related literature: minimizing distributional distance between embeddings of treated and control units, and minimizing the ability of an adversary to predict from the embedding whether a unit is treated or control. They prove theoretical results similar to those in Shalit et al (2017) connecting these two properties to treatment effect estimation error (e.g. PEHE). Their proposed method includes two types of embeddings corresponding to these two properties, and concatenates them for the full model (DIGNet). They show that DIGNet outperforms baseline approaches on standard treatment effect estimation semi-synthetic setups like IHDP and Twins.

**Audience:**

Yes

**Broader Impact Concerns:**

no concerns

**Claims And Evidence:**

No

**Requested Changes:**

- my main point is around the method - it's not clear to me how the distributional distance minimizing method and the adversarial optimization method are either different, or complementary. This is key and I wouldn't accept this paper without understanding this
- I would want to see more clarity in the intro around distribution shifts and the role of covariate shift in the scope
- the motivating example on p2 needs clarification I think - this may help quite a bit with overall motivation
- I'd like some extra thought put into empirical clarifications of a) why this approach works or b) why alternative explanations for the method working are not correct

**Strengths And Weaknesses:**

Strengths:
- empirical results are strong - demonstrates wins over baselines
- some useful theoretical results extending work of Shalit et al (2017) and Ben-David (2006)

Weaknesses & Suggestions:
- I find the proposed method somewhat confusing and the motivation a little unclear. It seems to me that the adversarial approach (minimizing treatment predictability) and the distributional discrepancy approach are somewhat redundant - it's been shown that one often implies the other, and so I don't see the benefit of concatenating both. Additionally, if in fact they do have different properties, it seems like concatenating them would be counterproductive; e.g. an adversary might perform poorly on the adversarially-optimized embedding, but well when we concatenate a non-adversarially-optimized embedding onto it. Because of this confusion I'm not able to gain much insight from the method or experimental results, which I feel like is particularly important in causal inference work; for instance, the theoretical results are nice but it's not so clear how the proposed setup helps optimize them
- I think there should be more clarity off the top (intro and exposition) around the role of covariate shift here. For instance, the term "covariate shift" is ellided with "selection bias" at the bottom of page 1 and seemingly general distribution shift as well, when in fact there are many types of distribution shift that can present issues. A discussion of this would be helpful, at least for scoping purposes
-I think the phrase "over-balancing" is not necessarily helpful - the issue is not always "too much" balancing, but rather the wrong kind of balancing when we lose accuracy
- the motivating example on p2 is unclear: it seems to me like in this case, if T=age, then mapping these representations onto each other perfectly would be helpful, since then we can build an estimator for T=old and T=young separately. I'm not sure I understand what the issue is in this case.
- the lack of insight around why this method might work makes me wonder if there is another reason, such as increased representational capacity achieved by concatenating representations. It would be good to either address some of these alternative reasons or clarify the motivations more

Smaller points:
- in 2.1, not sure why \phi needs to be invertible - I know that's true in some works but it doesn't in general need to be the case for learning balanced representations right?
- in 3, could use just a short explanation of what A and \sigma^2 are intended to be - I understand but would help with readability
- on bottom on p8, the authors claim that INet is novel - however there is quite a bit of work in the fair representation learning literature proposing very similar ideas, so I'm not sure I would agree with the claim. See "Censoring Representations with an Adversary" (Edwards & Storkey) and "Learning Adversarially Fair and Transferable Representations" (Madras et al) for two examples (should probably touch on this line of work in the related work section as well)
- could use some more information in 5.1 on how IHDP is created and which treated samples are removed to create selection bias
- I think the visualization in Fig 5 could be a bit stronger - no real reason AFAICT to lay out the x-axis over seeds like this. Also, would DGNet and DINet be useful to show in 5c and 5d respectively?

---

### Review · Reviewer_y2MN · 2024-03-11

**Summary Of Contributions:**

The paper proposes a new deep learning method and some theoretical bounds for the problem of individual treatment effect estimation.

**Audience:**

No

**Broader Impact Concerns:**

N.A.

**Claims And Evidence:**

Yes

**Requested Changes:**

I think the writing of the paper can be improved. Many sentences are hard to read and follow.

**Strengths And Weaknesses:**

Strengths:
1. The idea of the proposed method is easy to follow
2. The related works have been reviewed thoroughly
3. The simulation results are intensive and evident

Weaknesses:
1. The theory is incremental given the existing works in the literature.
2. I don't think the theory is sufficient to explain the improvements made by the method.

---

### Decision · Action_Editor_XNki · 2024-05-15

**Recommendation:** Accept with minor revision

**Comment:**

In their final assessments, two reviewers voted "leaning reject" and one voted "leaning accept".  The core issues for the two who voted leaning reject were in the motivation of the approach and the associated clarity of how this is presented in the paper.

Also factoring into my decision is that this is a "major revision" of a previous TMLR submission (https://openreview.net/forum?id=uyp8eFbzzT) that was only relatively narrowly rejected with clear requested changes. Unfortunately, it was only possible to re-recruit one of the same reviewers from the previous submission (they changed their assessment from leaning reject to leaning accept). I generally feel that most of the issues raised from the previous submission have been corrected, with the exception of the overall clarity of the work (especially in motivating its core approach) which I still believe is problematic.  I do not feel that many substantial new issues have been raised in this revision round, but the two new revisions took a more critical stance on the clarity issues with the paper and lack of sufficient motivation for the core approach.

Overall, I think this is still quite a borderline decision.  However, ultimately I do believe that the vast majority of the core issues in the paper have been addressed over the two submission cycles and that the paper is now suitable for publication at TMLR.  In particular, while I think that many justified criticisms have been raised by reviewers that would likely prohibit publication at top ML conference venues, I do think it satisfies the criteria set out for acceptance by TMLR.  My recommendation is therefore that the paper should now be accepted.

That said, I still think the paper has noticeable room for improvement in the clarity of its key ideas.  I would therefore like to request minor changes based on further refinement of the introduction and general exposition of the core contributions.  I would like to see the current introduction made significantly more concise (potentially moving some content elsewhere if needed) so that it provides a more direct and focussed motivation and summary of the work, and more clearly explains the rationale behind the precise representation concatenation being used (i.e. why the approach being used achieves the two principles introduced, rather than just motivating them in isolation).

**Audience:**

Though reviewers had concerns about clarity in the motivation of the introduced approach and whether it is likely to be useful, all believed that the paper meets TMLR's evaluation criteria for audience.  I agree with this consensus and believe that there will be people in TMLR's audience interested in the work, despite so reservations with the level of novelty and significance that might prohibit publication for other ML venues.

**Claims And Evidence:**

In the final assessment, two reviewers felt that the claims in the paper were sufficiently evidenced, while one did not feel this bar was quite reached (though they indicated that they felt that it was relatively borderline).  The primary concern here was based around the motivation of the approach, with two reviewers (and myself) feeling that the paper did not make a sufficient clear and concise case for the approach taken, in particular, why concatenating representations of different types is a useful idea in this space.

Revisions during the resubmit and rebuttal period have tried to address this, but I am not sure they have been entirely successful.  In particular, while the addition of lots of new content to the introduce has provide some more motivation in parts, it has not really addressed the question of why to take the precise DIGNet approach, focusing instead more of further motivation for the two principles laid out.  Moreover, the intro has now become very bloated, such that it is a bit unfocused and meandering: the reader now has a lot to go through the get a sense of the core points of the paper.  It is not clear to me that the intro itself has thus improved significantly in the revisions: it would be better to have more concise and direct motivation of the work, with some extra context and examples covered a bit later.